# Dual functions of SNAP25 in mouse taste buds

Kengo Horie[1,2] , Kuanyu Wang[1], Hai Huang[1], Keiko Yasumatsu[3], Yuzo Ninomiya[1,4,5] ,
Yoshihiro Mitoh[1,2] and Ryusuke Yoshida[1,2]

[1] *Department of Oral Physiology, Graduate School of Medicine, Dentistry and Pharmaceutical Sciences, Okayama University, Okayama, Japan*
[2] *Faculty of Medicine, Dentistry and Pharmaceutical Sciences, Okayama University, Okayama, Japan*
[3] *Tokyo Dental Junior College, Tokyo, Japan*
[4] *Graduate School of Dental Science, Kyushu University, Fukuoka, Japan*
[5] *Monell Chemical Senses Center, Philadelphia, PA, USA*

The peer review history is available in the Supporting Information section of this article
(https://doi.org/10.1113/JP288683#support-information-section).

**

**Abstract figure legend** Conditional knockout of *Snap25* in taste cells leads to a marked reduction in the number of sour-responsive Type III cells in both fungiform and circumvallete papillae of mice. In behavioural assays, *Snap25/Trpv1* double knockout mice display increased licking to sour tastants, suggesting impaired aversive responses. Furthermore, electrophysiological recordings from the chorda tympani nerve show diminished responses to sour stimuli in *Snap25* conditional knockout mice. Togheter, these results indicate that SNAP25 is essential not only for sour taste signal transmission but also for the maintenance of sour taste cells.

**Kengo Horie** earned his BS in Agricultural Science from Tohoku University in Japan, and his MS and PhD in Molecular Biology from the Graduate School of Tohoku University. He completed his postdoctoral training at the Centre for Translational Social Neuroscience at Emory University under the supervision of Dr Larry J. Young. He was subsequently appointed as an Assistant Professor in the Departments of Medicine, Pharmacology and Dental Science at Okayama University.

**Abstract** Type III cells in mouse taste buds are considered to transmit aversive stimuli, such as sourness, to the gustatory nerve through vesicular synapses. Synaptosome-associated protein 25 (SNAP25) might contribute to synaptic vesicular release in sour sensation, although direct evidence has been lacking. Here, we demonstrated that epithelia-specific *Snap25* conditional knockout (cKO) mice exhibited a significant reduction in the number of type III cells. Notably, the proportion of 5-ethynyl 2′-deoxyuridine-positive post-mitotic type III cells in *Snap25* cKO mice was significantly lower on tracing day 14, but not at day 7, which suggests that SNAP25 contributes to the maintenance of type III cells. In a short-term lick test, *Snap25* cKO (sour taste absent) and *Snap25*/ transient receptor potential vanilloid 1 double KO (sour taste and somatosensory absent) mice exhibit a significantly higher lick response to sour tastants, confirming the role of SNAP25 for sour sensation. Electrophysiological recordings of the chorda tympani nerve reveal nearly abolished ammonium and sour taste responses in *Snap25* cKO mice, which concludes sour-dependent synapse transmission in type III cells. Overall, these data suggest that vesicular synapses in taste buds are indispensable for transmission of information from, and the replenishment of, sour-sensitive type III taste cells.

(Received 3 February 2025; accepted after revision 6 May 2025; first published online 23 June 2025)

**Corresponding author** R. Yoshida: Department of Oral Physiology, Graduate School of Medicine, Dentistry and Pharmaceutical Sciences, Okayama University, Okayama 700-8525, Japan. Email: yoshida.ryusuke@okayama-u.ac.jp

### Key points

- The lack of SNAP25 in sour taste cells (*Snap25* conditional knockout) significantly reduces the number of sour taste cells in the fungiform and circumvallate papillae of mice.
- *Snap25/Trpv1* double knockout mice exhibit increased lick responses to sour tastants in the short-term lick test.
- The chorda tympani nerve of *Snap25* conditional knockout mice shows reduced electrophysiological responses to sour tastants.
- These findings demonstrate that SNAP25 contributes not only to sour taste signal transduction, but also to the maintenance of sour taste cells, underscoring the dual functions of SNAP25 in taste cells.

## Introduction

Taste sensation is a complex and fundamental process that enables organisms to detect and respond to a wide variety of chemical stimuli. The core of this system comprises the taste cells, which are categorized into four distinct cell types: type I glia-like cells; type II sweet, umami, bitter and amiloride sensitive salt-responsive cells; type III sour-responsive cells; and type IV basal stem cells (Castillo-Azofeifa et al., 2017; Kinnamon & Finger, 2022). Of these, type III cells are considered to transmit sour signals to the taste nerve via canonical vesicular synaptic transmission, a mechanism that is well-studied in the central nervous system but still elusive in taste buds (Sudhof, 2004).

To transmit signal at the synapse, neurons in the central nervous system recruit the SNARE complex, composed of Synaptobrevin2 (*Syb2*), Syntaxin1 (*Syn1*) and *Snap25* (Mishima et al., 2014; Schoch et al., 2001;

Söllner et al., 1993; Sudhof, 2004; Washbourne et al., 2002). SYB2, also known as VAMP2, is a vesicular SNARE protein (v-SNARE) anchored to the surface of a synaptic vesicle (Baumert et al., 1989). SYN1 and SNAP25 are target cell-associated SNARE proteins (t-SNARE) localized on the plasma membrane (Söllner et al., 1993). v-SNARE and t-SNARE proteins form a SNARE complex, triggering membrane fusion in a calcium-dependent manner, leading to the exocytosis of synaptic vesicles (Jahn & Scheller, 2006). The mechanism of vesicular synaptic transmission in type III cells is believed to be similar to that in the central nervous system because they partly share an expression profile of synapse-related genes (Sukumaran et al., 2017; Yang et al., 2000, 2004, 2007). Of these synapse related proteins, the localization of SNAP25 has been well characterized in mouse taste buds. SNAP25 is exclusively localized in type III taste cells, as demonstrated by immunohistochemical studies in mice with or without genetic modification (Clapp et al., 2006;

Lossow et al., 2017; Tomchik et al., 2007). Although these observations support the idea that the SNARE complex is the main component in transmitting sour signals from type III cells, whether SNAP25 directly contributes to sour taste transmission has yet to be experimentally demonstrated.

By contrast, recent studies on type II cells have clearly demonstrated that a channel synapse, a form of novel synaptic transmission mediated by voltage-dependent ATP channels (CALHM1/3), transmits sweet, umami, bitter and salt signals from type II cells to taste nerves (Taruno et al., 2021). CALHM1/3 mediates the release of ATP from the cytosol into the synaptic cleft (Ma et al., 2018; Taruno et al., 2013). ATP then activates P2X receptors on the postsynaptic terminal of taste nerves, evoking action potentials and thereby conveying taste signals to the solitary nucleus in the central nervous system (Finger et al., 2005). On the other hand, type III cells are thought to release some neurotransmitters to convey sour taste signals to the peripheral gustatory nerve. Among the identified potential neurotransmitters, ATP and 5-hydroxytryptamine (5-HT, serotonin) play key roles in this process. ATP, acting on P2X receptors, is essential for activating the peripheral nerve, whereas 5-HT partially contributes to afferent nerve responses to sour stimuli (Finger et al., 2005; Huang et al., 2005; Kinnamon & Finger, 2013). However, type III cells do not express vesicular ATP transporters, which are responsible for loading ATP into synaptic vesicles in neurons (Iwatsuki et al., 2009). This absence leaves the exact role of synaptic vesicular release in activating nerve fibres in response to sour stimuli unclear. Understanding the differences in the mechanisms of synaptic transmission between type II and III cells will ultimately lead to a deeper interpretation of the complex synaptic system, not only in taste buds, but also in other synapse-forming peripheral organs.

In the present study, aiming to reveal the contribution of a canonical vesicular synaptic system for sour taste signal transmission in type III cells, we generated epithelia-specific *Snap25* conditional knockout mice (*Snap25* cKO) by crossing *Keratin5 Cre* (*K5^Cre*) and *Snap25^flox/flox* (*Snap25-floxed*) mice (Gustus et al., 2018; Tarutani et al., 1997). We examined whether deletion of *Snap25* influences the profile of the taste cell types using immunohistochemistry and 5-ethynyl 2′-deoxyuridine (EdU) cell proliferation tracking. We also assessed short-term licking behaviour in *Snap25* cKO mice, with or without *transient receptor potential vanilloid 1* (*Trpv1*), a cation channel expressing in the trigeminal nerve that is activated by extracellular proton, to evaluate the necessity of *Snap25* in taste behaviours. Finally, we recorded electrophysiological responses of the chorda tympani (CT) nerve to demonstrate the role of *Snap25* in synaptic transmission.

## Methods

### Animals

All animal procedures were performed in accordance with protocols approved by the Animal Committee at Okayama University (Approved Nos: 2022860, 2022305, 2021811, 2021835, 2021812, and 2024290). The *K5^Cre* mouse line, which produces Cre recombinase under the control of K5 promoter, was originally developed and imported from Osaka University (Dr Junji Takeda) (Tarutani et al., 1997). The *Snap25-floxed* mouse line was developed and imported from the University of New Mexico (Drs Michael C. Wilson and Lee Anna Cunningham) (Gustus et al., 2018). *Trpv1* KO mice were derived from the University of California and imported from the National Institute for Physiological Sciences, Japan (Drs David Julius and Makoto Tominaga) (Caterina et al., 1997). Both male and female mice were used in all experiments, with no observed sex differences in the results (data not shown). Animals aged between 10 and 24 weeks old were housed under a 12:12 h light/dark photocycle at 23°C with access to food and water available *ad libitum*.

### Immunostaining

Immunostaining was performed as previously described (Niki et al., 2015; Yoshida et al., 2015). Briefly, animals were killed with $CO_2$. For fungiform papillae (FP), the anterior part of the tongue was removed from the tongue and 0.25 mg mL$^{-1}$ elastase in Tyrode's solution was injected between the epithelium and muscle. The epithelium was peeled off and fixed with 4% paraformaldehyde (PFA), which was then used for subsequent immunostaining. For circumvallate papillae (CV), the posterior portion of the tongue was dissected and fixed in 4% PFA for 45 min at 4°C. After dehydration in a sucrose solution, tissues were embedded in OCT compound (Sakura Finetek, Tokyo, Japan) and sectioned into 10 μm thick sections, which were mounted on silane-coated glass slides. Slides were washed with Tris-buffered saline (TBS), incubated in Blocking One-P (Nacalai tesque, Kyoto, Japan) for 1 h at room temperature and incubated overnight at 4°C with the primary antibodies: anti-carbonic anhydrase 4 (CA4) (goat-IgG; dilution 1:400, AF2414; R&D Systems, Minneapolis, MN, USA; RRID = AB_2 070 332), SNAP25 (rabbit-IgG; dilution 1:100; ab41455; Abcam, Cambridge, UK; RRID = AB_945 552), gustducin (GUST) (goat-IgG; dilution 1:200; Aviva Systems Biology, San Diego, CA, USA; RRID = AB_10 882 823) or ectonucleoside triphosphate diphosphohydrolase 2 (ENTPD2) (sheep IgG; dilution 1:400; AF5797; R&D Systems; RRID = AB_10 572 702). After washing with

TBS, the slides were incubated with secondary antibodies: anti-rabbit IgG (Alexa Fluor 488 donkey anti-rabbit IgG H+L; dilution 1:200; Thermo Fisher Scientific, Waltham, MA, USA; RRID = AB_2 535 792), goat IgG (Alexa Fluor 568 donkey anti-goat IgG H+L; dilution 1:200; Thermo Fisher Scientific; RRID = AB_2 534 104), or sheep IgG (donkey anti-sheep IgG H&L Alexa Fluor 568; dilution 1:200; Abcam; RRID = AB_2 892 984). Fluorescence signals were visualized using a laser scanning microscope (LSM780; Zeiss, Oberkochen, Germany; or FX-300; Olympus, Tokyo, Japan) and analysed with Zen software (Zeiss) or FLUOVIEW (Olympus).

### EdU cell proliferation tracking

Mice were I.P. injected with 5 mg kg$^{-1}$ body weight of EdU once a day at days 1–3. On days 10 or 17 (7 or 14 days after the last injection), the mice were killed, and their tongue epithelium containing FP was enzymatically removed and fixed with 4% PFA (as in the immuno-staining procedure). After washing with TBS and blocking with Blocking One-P, tissues were treated with a Click-iT EdU Cell Proliferation Kit for Imaging, Alexa Fluor 488 dye (Thermo Fisher Scientific) in accordance with the manufacturer's instruction. Samples were then incubated overnight at 4°C with anti-CA4 (RRID = AB_2 070 332) or GUST (RRID = AB_10 882 823) antibodies, followed by incubation with secondary antibodies: anti-goat IgG Alexa Fluor 568 (RRID = AB_2 534 104) antibody. Fluorescence signals were visualized and analysed as described above.

### Whole nerve electrophysiological recording of the CT nerve

Whole nerve electrophysiological responses of the CT nerve were recorded as previously described with modifications in anaesthesia, recording and analysis (Kusuhara et al., 2013; Mikami et al., 2024). Mice were anesthetized with an I.P. injection of a combination of 0.3 mg kg$^{-1}$ medetomidine, 4.0 mg kg$^{-1}$ midazolam and 5.0 mg kg$^{-1}$ butorphanol, with supplemental half-dose injections every 2 h to maintain anaesthesia. Depth of anaesthesia was confirmed by a toe pinch with tweezers. The right CT nerve was exposed by removing the pterygoid muscle and cut to a sufficient length to place it on an Ag–AgCl electrode. Approximately 1 mL of each solution was applied to the tongue via a 5 mL syringe. The solutions used were: 100 mM NH$_4$Cl (02424-55; Nacalai tesque), 20 mM saccharin (Sacc) (197-08605; FUJIFILM, Tokyo, Japan), 10–1000 mM sucrose (Suc) (30404-05; Nacalai tesque), 10–1000 mM NaCl, 10–300 mM mono-potassium glutamate (MPG) (A17232.30; Thermo Fisher Scientific), 0.1–20 mM quinine-HCl (QHCl) (29910-32;

Nacalai tesque), 0.3–30 mM hydrochloric acid (HCl) (37314-15; Nacalai tesque), 0.3–30 mM citric acid (CA) (09109-85; Nacalai tesque), 30 mM acetic acid (AA) (017-00256; FUJIFILM), 30 mM lactic acid (LA) (Musashino, Tokyo, Japan) and 30 mM tartaric acid (TA) (32702-75; Nacalai tesque). Taste responses were recorded for 30 s at intervals of 1 min after rinsing with distilled water (DW). Neural activity was amplified by DAM80 (World Precision Instruments, Sarasota, FL, USA) and monitored on an oscilloscope. Integrated whole nerve responses (time constant: 1.0 s) were recorded by PowerLab software (ADInstruments, Bella Vista, NSW, Australia). The middle 10 s of each 30 s response were extracted to remove noise of tactile responses from the onset and offset of taste stimulation. All responses were normalized to baseline and presented as a ratio to responses to 20 mM Sacc (= 1 normalized response). Surgery and recording procedures lasted 2–4 h and animals were killed with CO$_2$ immediately after the experiment.

### Short-term lick test

A short-term lick test was performed as previously described (Yamase et al., 2023). Briefly, singly housed animals were water-deprived for 12 h and trained to lick DW in 5 s periods followed by 10 s intervals for 5 days. From day 6, the number of licks to a test solution or DW was counted using a lick meter (Yutaka Electronics Co., Gifu, Japan). The solutions tested were: 1–100 mM HCl, 1–100 mM CA, 30–1000 mM Suc, 10–300 mM mono-sodium glutamate (MSG) (16914-05; Nacalai tesque), 0.01–1 mM QHCl, 30–1000 mM NaCl and 1–1000 mM potassium chloride (KCl) (28513-85; Nacalai tesque). One tastant, at varying concentrations, was tested on any given test day. To examine lick responses to preferred solutions (Suc and MSG), mice were deprived of both food and water 12 h before the experiment. On each test day, mice were given test solutions with concentrations of descending order (from highest concentration to DW) in the first trial then randomized order in subsequent trials. To examine lick responses to aversive solutions (NaCl, QHCl, CA and HCl), mice were deprived of water 12 h before the beginning of experiment. On each test day, mice were given test solutions in ascending concentration order (from DW to highest concentration) in the first trial and then randomized order for subsequent trials. The number of lick trials for each solution was at least three, and their values were averaged for data analysis.

### Statistical analysis

For immunostaining and EdU experiments, the number of immunoreactive cells was manually counted and

statistically compared using Student's *t* test or repeated two-way ANOVA followed by a *post hoc* Tukey's test. For the short-term lick test, repeated two-way ANOVA was performed to compare the concentration of each solution, genotype and interaction effects, and Tukey's honestly significant difference (HSD) was followed as *post hoc* analysis. For the CT nerve recording, repeated measured two-way ANOVA was performed for analysing sucrose, NaCl, MPG, QHCl, HCl and CA. Sidak's multiple test was performed when applied. $P < 0.05$ was considered statistically significant. The data in the figure 5 and 6 were presented as mean $\pm$ SD.

## Results

### Generation of *Snap25* cKO mice

In the taste system, *Snap25* is exclusively expressed in type III cells among adult taste cells and peripheral nerve fibres in mice (Wilson et al., 2017). To specifically analyse the functions of SNAP25 in taste cells (type III cells), we first generated epithelia-specific *Snap25* cKO mice by crossing $K5^{Cre}$ and *Snpa25-floxed* mice. *K5* is expressed in the basal layer of epithelium in the tongue, but not in the peripheral neurons, and the cells expressing *K5* develop into all types of adult taste cells (Okubo et al., 2009). Thus, *Snap25* cKO mice were expected to lack SNAP25 proteins in type III cells of the taste buds. We confirmed that SNAP25 immunoreactivity was completely undetectable in taste cells in FP (Fig. 1*A*) and CV (Fig. 1*B*) of *Snap25* cKO mice, although that in the peripheral nerve remained detectable. Noteworthy, although whole-body deletion of *Snap25* gene causes postnatal death as a result of the malfunction of respiratory system in mice (Washbourne et al., 2002), *Snap25* cKO mice evaded lethality and were analysed in the subsequent experiments.

### Type III cells were reduced in the taste buds of *Snap25* cKO mice

Representative image sets from a single animal of each genotype visually demonstrated that the conditional ablation of *Snap25 m*ay reduce the number of CA4-positive type III cells, but not GUST-positive type II cells (Fig. 1*C*). To examine whether taste cells were normally retained in the taste buds of *Snap25* cKO mice, we quantified the number of GUST-positive cells as type II cells and CA4-positive cells as type III cells in FP and CV of *Snap25*-floxed and cKO mice. Because the morphology of type I cells is ambiguous and technically difficult to quantify, type I cells were not counted in this study, but were visually assessed. GUST-positive type II cells of *Snap25* cKO mice were normally retained compared to those of *Snap25*-floxed mice in both FP

($P = 0.266$) (Fig. 2*A*) and CV ($P = 0.208$) (Fig. 2*B*). However, the number of CA4-positive type III cells of *Snap25* cKO mice was significantly lower than that in *Snap25*-floxed mice in both FP ($P < 0.001$) (Fig. 2*C*) and CV ($P < 0.001$) (Fig. 2*D*). These results demonstrate that SNAP25 is essential for the cell survival of type III cells in the adult taste buds of mice.

### Type III cells were normally replenished but not maintained in FP of *Snap25* cKO mice

This result raised the question of whether the turnover of taste cells in *Snap25* cKO is maintained. The turnover of mouse taste cells depends on their cell type (Perea-Martinez et al., 2013). The half-life of type III cells is 22 days, and some cells are maintained for over 40 days in mouse taste buds. We hypothesized that the reduction of type III cells in *Snap25* cKO mice could be the result of a reduction in (1) the replenishment or (2) the maintenance of type III cells. To test these hypotheses, post-mitotic type III cells in live animals were labelled with EdU and quantified by counting EdU and CA4 double-positive cells in FP on days 7 and 14 after the onset of labelling (Fig. 3*A–D*). As a control for type III cells, EdU and GUST double-positive type II cells were quantified under the same time course. Visual comparison of representative image sets from a single animal under each genotype and condition indicated a potential reduction in EdU and CA4 double-positive type III cells in the absence of SNAP25, without affecting EdU and GUST double-positive type II cells (Fig. 3*E*). These representative observations motivated us to further quantitatively analyse the number of post-mitotic cells. The number of labelled type II cells was consistent between days 7 and 14 in both genotypes (Fig. 4*A* and Table 1). A two-way ANOVA revealed significant main effects of Genotype and Treatment day on the number of EdU and CA4 double-positive cells per taste bud, with no significant interaction between Genotype and Treatment day (Fig. 4*B* and Table 1). In addition, the proportion of EdU and CA4 double-positive cells relative to total CA4-positive cells was significantly lower in *Snap25* cKO mice on day 14 compared to day 7, although this difference was not significant in *Snap25*-floxed mice (Fig. 4*C* and Table 1). Notably, the increase in the proportion of EdU and CA4 double-positive cells in *Snap25* cKO mice on day 7 suggests that surviving post-mitotic type III cells were concentrated because of the reduction in total type III cells. On day 14, the loss of EdU and CA4 double-positive cells further decreased their proportion, indicating that type III cells are no longer maintained (Fig. 4*C* and Table 1). Taken together, these findings suggest that type III cells are normally replenished and maintained for at

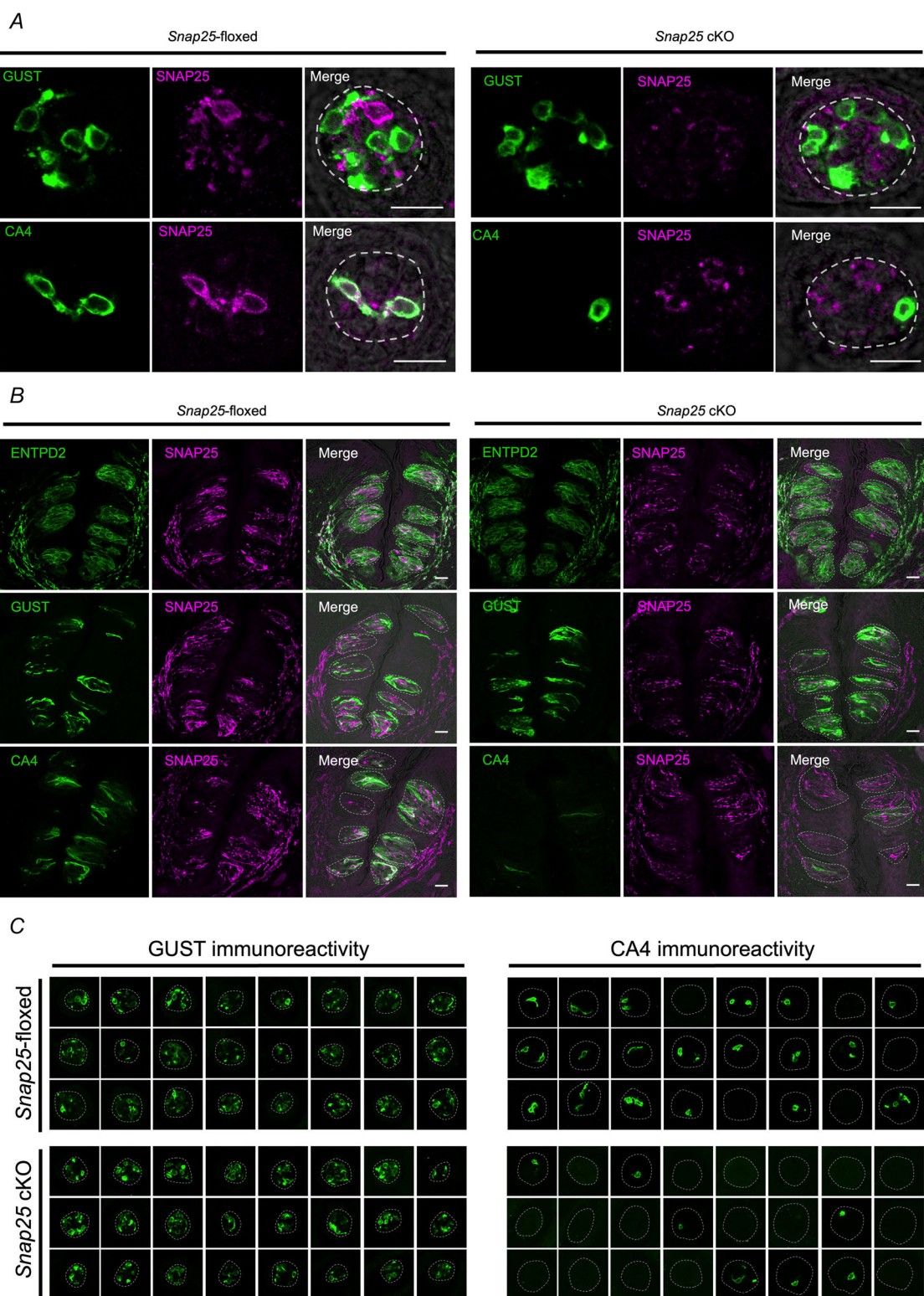

**Figure 1. Immunostaining of types II and III cell markers in the FP and CV of *Snap25-floxed and* cKO mice**

*A*) immunoreactivities of taste cell markers (GUST and CA4) and SNPA25 in fungiform papillae (FP) of *Snap25*-floxed (left) and *Snap25* cKO mice (right). *B*) immunoreactivities of taste cell markers (ENTPDs, GUST and CA4) and SNPA25 in circumvallate papillae (CV) of *Snap25*-floxed (left) and *Snap25* cKO mice (right). Green: immuno-reactivity of ENTPD2, Gust or CA4. Magenta: immunoreactivity of SNAP25. Scale bar = 20 μm. *C*) typical examples

of immunohistochemical images for GUST (left) and CA4 (right) in fungiform papillae of a *Snap25*-floxed (upper) and *Snap25* cKO mouse (lower). Twenty-four images of fungiform papillae were obtained from an animal of each genotype. Taste buds are outlined with dotted lines. *N* = 3 animals per group. [Colour figure can be viewed at wileyonlinelibrary.com]

**Table 1. Summary of statistical analyses related to Fig. 4**

| Figure no. | | Statistical method | *P* value |
|---|---|---|---|
| 4*A* | Two-way ANOVA | Genotype | 0.271 |
| | | Treatment day | 0.672 |
| | | Genotype × Treatment day interaction | 0.702 |
| 4*B* | Two-way ANOVA | Genotype | 0.0162 |
| | | Treatment day | < 0.001 |
| | | Genotype × Treatment day interaction | 0.0770 |
| 4*C* | Two-way ANOVA | Genotype | < 0.001 |
| | | Treatment day | < 0.001 |
| | | Genotype × Treatment day interaction | < 0.001 |
| | Tukey's HSD | *Snap25*-floxed day 7 *vs*. day 14 | 0.957 |
| | | *Snap25* cKO day 7 *vs*. day 14 | < 0.001 |

least 7 days, but are no longer maintained after 14 days of turnover in the FP of *Snap25* cKO mice.

### *Snap25* cKO mice demonstrated a reduction in sour responses of the CT nerve

SNAP25 has been hypothesized to be a main contributor of synaptic vesicular release from type III cells to peripheral nerves at the synapse in response to sour stimuli, although direct evidence from genetically modified animals has not yet been presented. To test this, we then analysed the electrophysiological responses of the CT nerve in *Snap25*-floxed and cKO mice (Fig. 5*A*). Although previous reports normalized the taste responses to 100 mM NH$_4$Cl responses (Niki et al., 2015; Yoshida et al., 2010),

the nerve responses to each taste solution were normalized to 20 mM Sacc because 100 mM NH$_4$Cl responses in *Snap25* cKO mice were significantly reduced compared to those in *Snap25*-floxed mice (*P* < 0.001) (Fig. 5*B*). CT nerve responses to 30 mM HCl and CA in *Snap25* cKO mice were significantly impaired compared to those in *Snap25*-floxed control animals (Fig. 5*C* and *D* and Table 2). CT nerve responses to various concentration of Suc, MPG, QHCl, and NaCl were not affected by genotype (Fig. 5*E–H* and Table 2). In line with HCl and CA responses, CT nerve responses to 30 mM AA, LA and TA in *Snap25* cKO mice were significantly diminished compared to those in *Snap25*-floxed control animals (AA, *P* = 0.0072; LA, *P* = 0.0015; TA, *P* = 0.0062) (Fig. 5*I–K*). Altogether, SNAP25 is a critical synaptic protein that

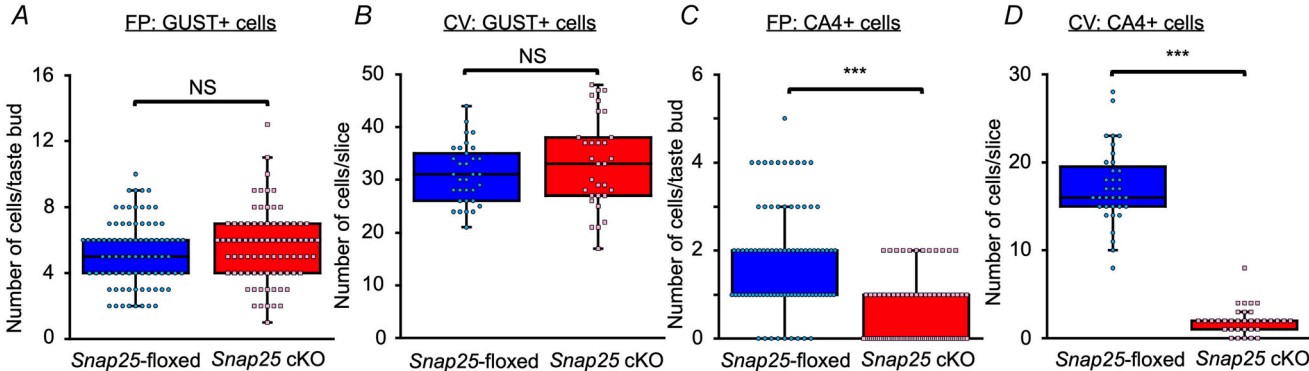

**Figure 2. Numbers of type III cells were less in *Snap25* cKO than in *Snap25*-floxed mice**
*A–D*) numbers of taste cells/taste bud or section in *Snap25*-floxed and *Snap25* cKO mice. Numbers of GUST+ cells *A* and *B*) or CA4+ cells *C* and *D*) in FP taste bud *A* and *C*) or CV slice *B* and *D*) in *Snap25*-floxed and *Snap25* cKO mice. The box indicates the 25th and 75th percentiles; the line across the box, the median; whiskers, maximum and minimum values. FP taste bud (*n* = 83–87), CV section (*n* = 31–35), animals (*n* = 3 per group). NS, not significant; ***P* < 0.001, Student's *t* test. [Colour figure can be viewed at wileyonlinelibrary.com]

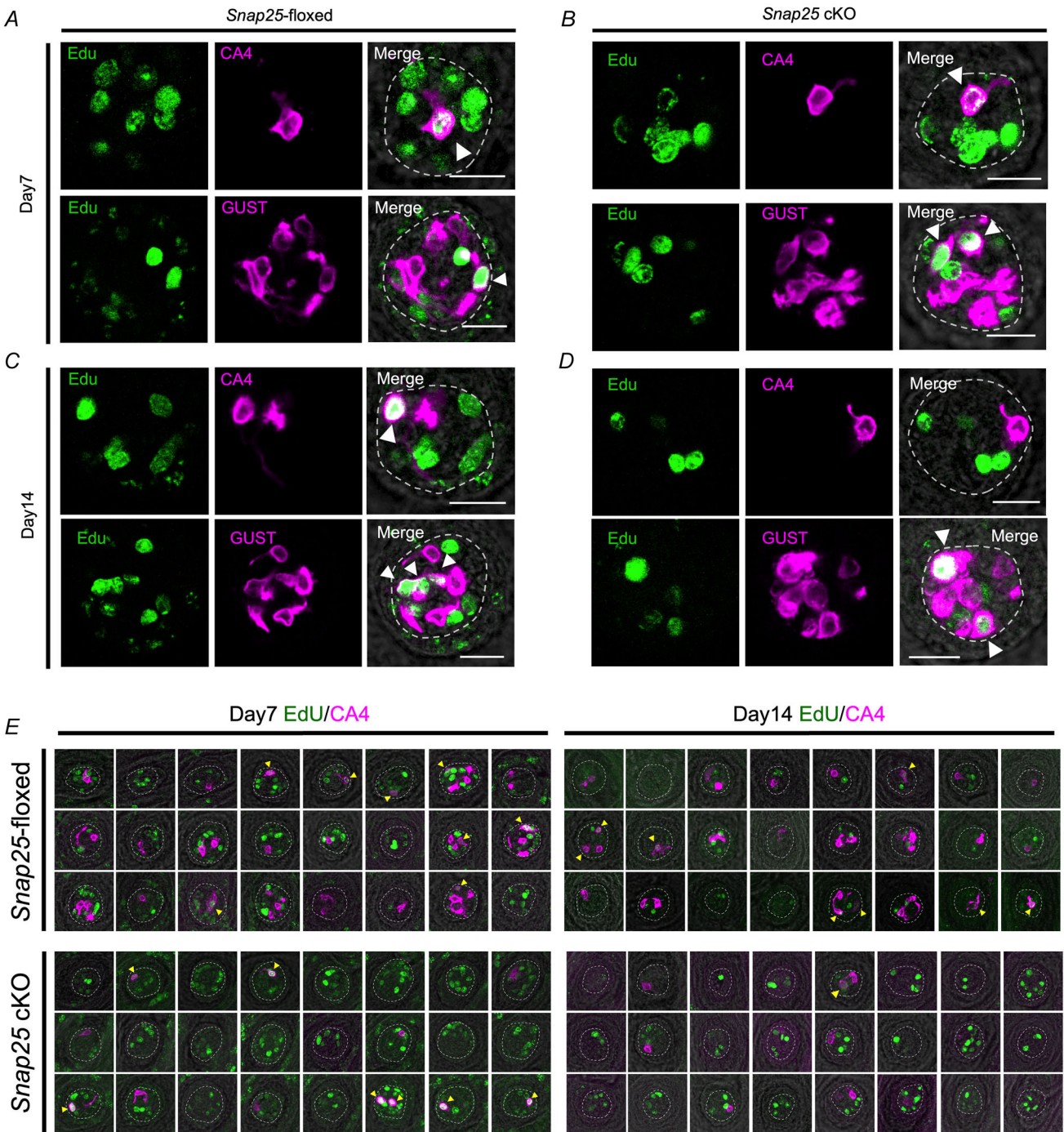

**Figure 3.  Double staining of EdU and GUST (type II cell marker) or CA4 (type III cell marker) in fungiform papillae of *Snap25*-floxed and cKO mice**
A–D) detection of EdU and taste cell markers in fungiform taste buds of *Snap25*-floxed A and C) and *Snap25* cKO mice B and D) 7 A and B) or 14 C and D) days after EdU administration. Green: fluorescence detection of EdU. Magenta: immunoreactivity of CA4 or GUST. Arrowheads indicate taste cells showing EdU staining and immunoreactivity of a taste cell marker. E) typical examples of immunohistochemical images for CA4 and EdU in fungiform papillae of *Snap25*-floxed (top) and cKO (bottom) mouse 7 days (left) and 14 days (right) after EdU administration. Twenty four images of fungiform papillae were obtained from a *Snap25*-floxed or cKO mouse for each condition. Arrowheads indicate taste cells showing EdU staining and immunoreactivity of CA4. Taste buds are outlined with dotted lines. Scale bar = 20 μm. [Colour figure can be viewed at wileyonlinelibrary.com]

contributes to sour signal transmission from type III taste cells to the peripheral taste nerve.

### *Snap25/Trpv1* double knockout mice exhibited a reduced aversion to sour tastants

To test whether SNAP25 in type III cells contributes to aversive licking responses to sour solutions in mice, we assessed short-term licking behaviour of control (*K5^Cre^*, *Snap25*-floxed, and *K5^Cre^*; *Trpv1* KO), *Snap25* cKO and *Snap25*; *Trpv1* double KO (DKO) mice. A previous study demonstrated that the single deletion of Otopetrin1 (*Otop1*), a proton-conducting channel functioning as a sour receptor in the taste buds, was insufficient to eliminate behavioural aversion to sour solutions in mice. However, *Otop1* KO mice lacking *Trpv1*-expressing trigeminal neurons exhibited impaired aversive licking responses to sour solutions (Zhang et al., 2019). This result led us to hypothesize that a single deletion of *Snap25* would be insufficient to induce a phenotype, but that *Snap25* and *Trpv1* DKO mice would exhibit it. As expected, a single deletion of *Snap25* (*Snap25* cKO)

resulted in aversive responses to various concentrations of HCl and CA solutions (Fig. 6*A* and *B* and Table 3). However, *Snap25*; *Trpv1* DKO mice showed significantly reduced aversive responses to 10 mM HCl and CA solutions (Fig. 6*A*, *B*, *H*, *I* and Table 3). Both *Snap25* cKO and *Snap25*; *Trpv1* DKO mice exhibited normal taste responses to Suc, MSG, QHCl, NaCl and KCl (Fig. 6*C–G* and Table 3). Aversive behavioural responses to 10 μM capsaicin in *Snap25* cKO mice were comparable to those of *K5^Cre^* and *Snap25*-floxed control animals (Fig. 6*J* and Table 3). suggesting that the trigeminal somatosensory system remains intact in *Snap25* cKO mice. Overall, SNAP25 contributes to the peripheral sour sensation in mice.

## Discussion

In taste buds, type II and type III cells express specific taste receptors and are responsible for detecting the five basic tastes (Taruno et al., 2021). Type II cells detect sweet, bitter, umami and amiloride-sensitive salt, whereas type III cells are responsible for detecting sour taste

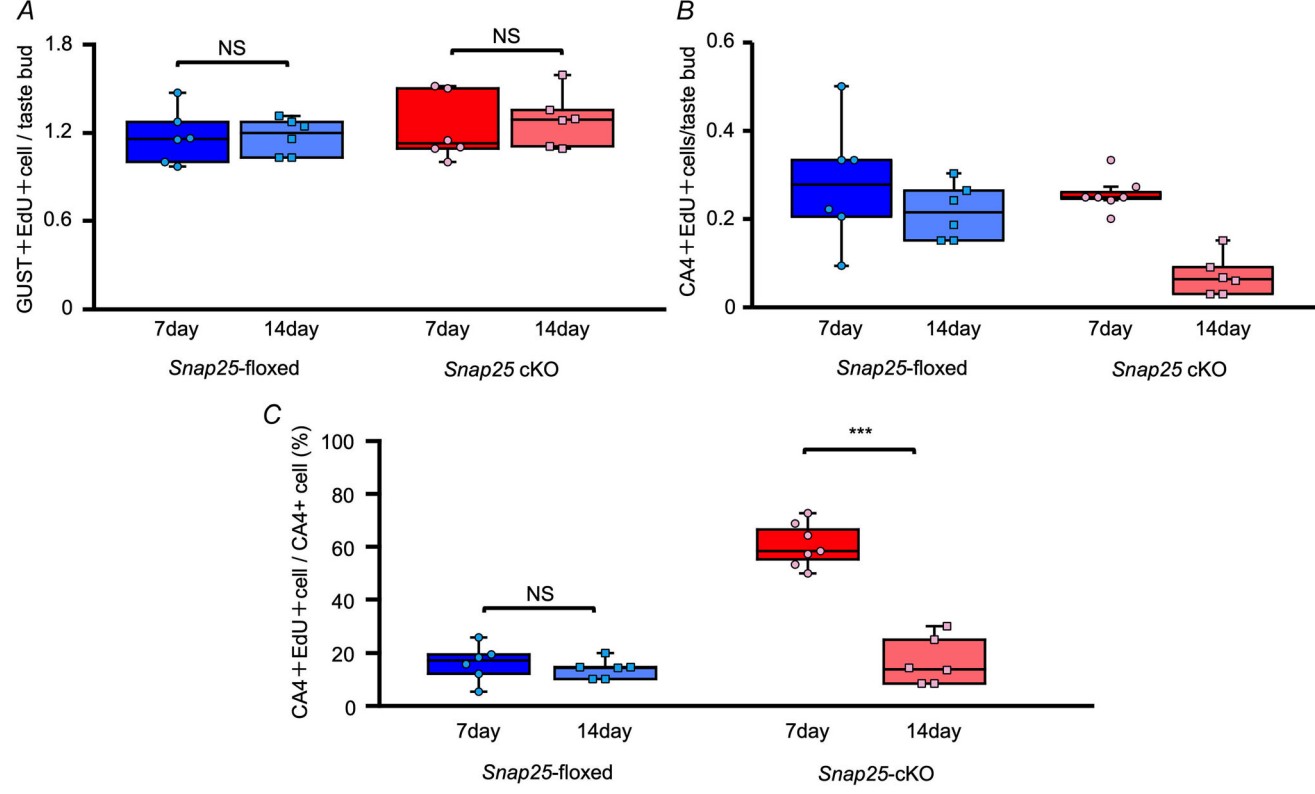

**Figure 4. Maintenance of type III cells was impaired in *Snap25* cKO mice**
*A* and *B*) comparison of double positive taste cells in taste buds between 7 days and 14 days after administration of EdU in *Snap25*-floxed and *Snap25* cKO mice. The box plots indicate the 25th and 75th percentiles; the line across the box, the median; whiskers, maximum and minimum values. *C*) the proportion of EdU and CA4 double-positive cells relative to total CA4 positive cells in *Snap25*-floxed and cKO mice. N = 6–7 animals (*n* = 24–35 FP taste buds in each animal). NS, not significant; ***$P < 0.001$, two-way ANOVA with a *post hoc* Tukey's test. [Colour figure can be viewed at wileyonlinelibrary.com]

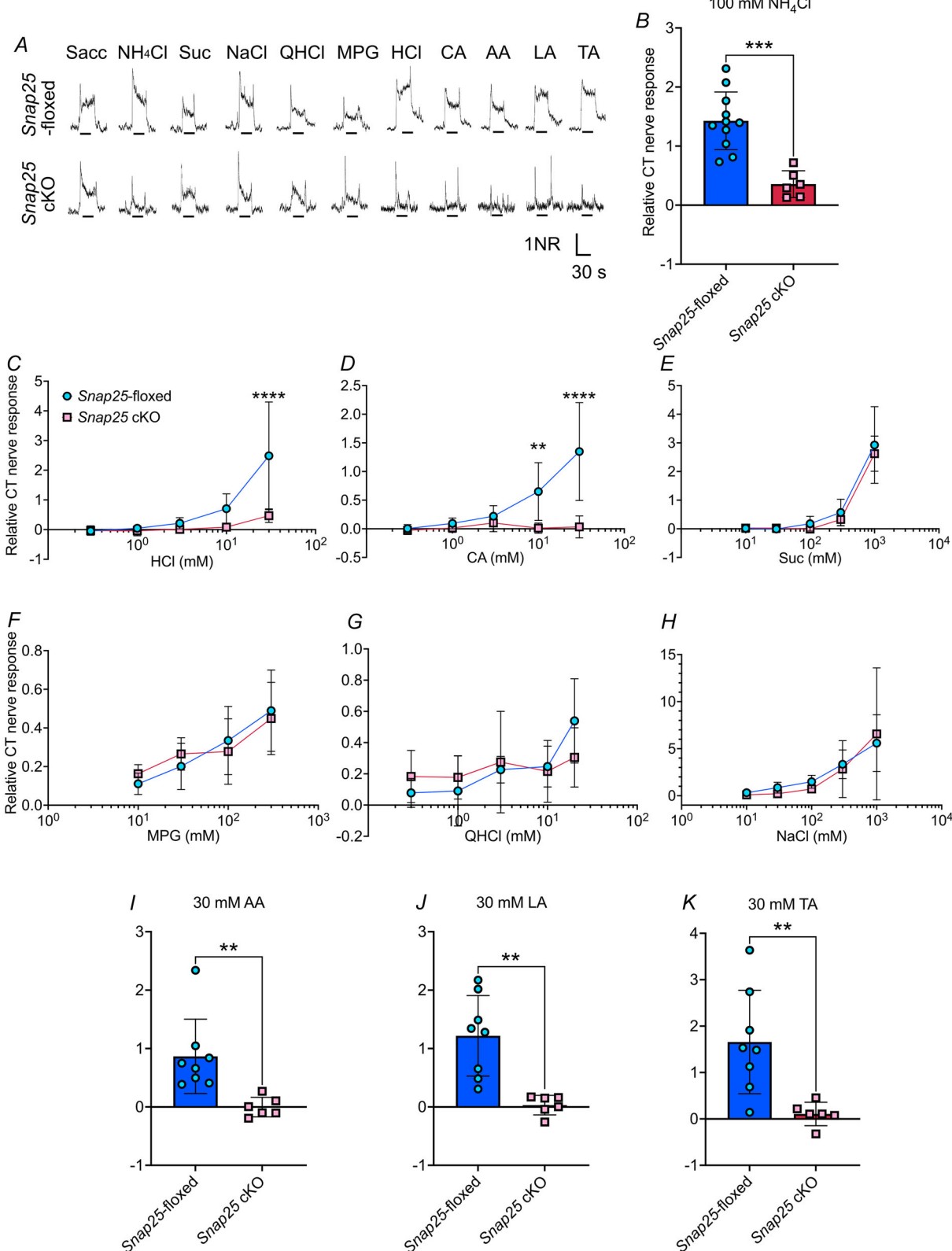

**Figure 5. CT nerve responses to ammonium and sour tastants were reduced in *Snap25* cKO mice**
*A*) representative CT nerve responses to 20 mм Sacc, 100 mм ammonium chloride (NH₄Cl), 300 mм Suc, 100 mм NaCl, 20 mм QHCl, 30 mм MPG, 30 mм hydrochloric acid (HCl), citric acid (CA), acetic acid (AA), lactic acid (LA) and tartaric acid (TA) in Snap25-floxed (closed circle) and Snap25 cKO (open circle) mice. Horizontal scale bar = 30 s, vertical scale bar = 1 normalized response (NR). *B*) CT nerve responses to 100 mм NH₄Cl in *Snap25*-floxed (*n* = 11)

and *Snap25* cKO mice ($n = 6$). ***$P < 0.001$ in $NH_4Cl$. Unpaired *t* test. CT nerve responses to various concentrations of HCl *C*), CA *D*), Suc *E*), MPG *F*), QHCl *G*) and NaCl *H*) in *Snap25*-floxed and *Snap25* cKO mice. ****$P < 0.0001$ in 30 mM HCl and CA. **$P < 0.01$ in 10 mM CA. Number of *Snap25*-floxed and *Snap25* cKO mice: HCl, $n = 6$–8; CA, $n = 6$–9; Suc, $n = 6$–8; MPG, $n = 6$–7; QHCl, $n = 6$–8; NaCl, $n = 6$–8. CT nerve responses to a single concentration of AA (*I*), LA (*J*) and TA (***K***) in *Snap25*-floxed and *Snap25* cKO mice. ***$P < 0.001$ in $NH_4Cl$, **$P < 0.01$ in AA ($n = 5$–7), LA ($n = 6$–8) and TA ($n = 6$–8). Unpaired *t* tests were performed. All data are presented as the mean $\pm$ SD. [Colour figure can be viewed at wileyonlinelibrary.com]

**Table 2. Summary of statistical analyses related to Fig. 5**

| Figure no. | | Statistical method | *P* value |
|---|---|---|---|
| 5*C* (HCl) | Two-way ANOVA | Genotype | 0.0121 |
| | | Concentration | < 0.001 |
| | | Genotype × Concentration interaction | < 0.001 |
| | Tukey's HSD | 30 mM HCl | < 0.001 |
| 5*D* (CA) | Two-way ANOVA | Genotype | 0.0042 |
| | | Concentration | < 0.001 |
| | | Genotype × Concentration interaction | < 0.001 |
| | Tukey's HSD | 10 mM CA | 0.0073 |
| | | 30 mM CA | < 0.001 |
| 5*E* (Suc) | Two-way ANOVA | Genotype | 0.264 |
| | | Concentration | < 0.001 |
| | | Genotype × Concentration interaction | 0.895 |
| 5*F* (MPG) | Two-way ANOVA | Genotype | 0.919 |
| | | Concentration | < 0.001 |
| | | Genotype × Concentration interaction | 0.472 |
| 5*G* (QHCl) | Two-way ANOVA | Genotype | 0.868 |
| | | Concentration | < 0.001 |
| | | Genotype × Concentration interaction | 0.0677 |
| 5*H* (NaCl) | Two-way ANOVA | Genotype | 0.793 |
| | | Concentration | < 0.001 |
| | | Genotype × Concentration interaction | 0.825 |

(Nomura et al., 2020; Tu et al., 2018; Zhang et al., 2003). These cell types employ distinct mechanisms for transmitting signals to the gustatory nerve: type II cells utilize CALHM1/3 mediated channel synapses, whereas type III cells use vesicular synapses (Taruno et al., 2013; Yang et al., 2000). Our findings highlight the presence of these two distinct signal transmission mechanisms in the peripheral taste system is essential for the transmission of sour taste, but not for the other basic tastes.

SNAP25 is a presynaptic t-SNARE component that primarily contributes to the exocytosis of synaptic vesicles in the central and peripheral nervous systems. A previous study reported that the deletion of *Snap25* in the inner hair cells induced deafness in mice as a result of reduced exocytosis, which was caused by degeneration of inner hair cells (Calvet et al., 2022). This degeneration could be induced by a deficit in plasma membrane recycling, as reported in cultured neurons *in vitro* (Peng et al., 2013). The similar elimination observed in type III cells in the taste buds of *Snap25* cKO mice suggests that SNAP25 could also be necessary for maintaining the cell survival through plasma membrane recycling in type III cells,

although further experiments are required to confirm this idea. In addition, the exclusive expression of *Snap25* in type III cells and type III cell-specific elimination in *Snap25* cKO mice (Figs 1–4) suggest that type II cells may recruit a SNAP25 independent mechanism for cell survival. Indeed, certain serotypes of botulinum neurotoxins specifically cleave SYN or SNAP25, causing cell degeneration in neurons, but not in different cell lines, such as neuroblastoma cell lines or primary glia cells (Peng et al., 2013). Additionally, cell degeneration caused by botulinum neurotoxin/C cleavage of SYN1 and SNAP25 can be rescued by the compensation of SYN2, 3 or 4 and SNAP23. These findings suggest the possibility of cell-type-specific combinations of t-SNARE proteins, which may explain differences in the longevity of each taste cell type.

The specific reduction in the CT nerve responses to sour tastants in *Snap25* cKO supports the labelled line theory between the taste buds and gustatory nerves (Roper & Chaudhari, 2017). Sour signals tested in this study were transmitted through synaptic vesicular release of neurotransmitters, such as ATP or serotonin, from type III

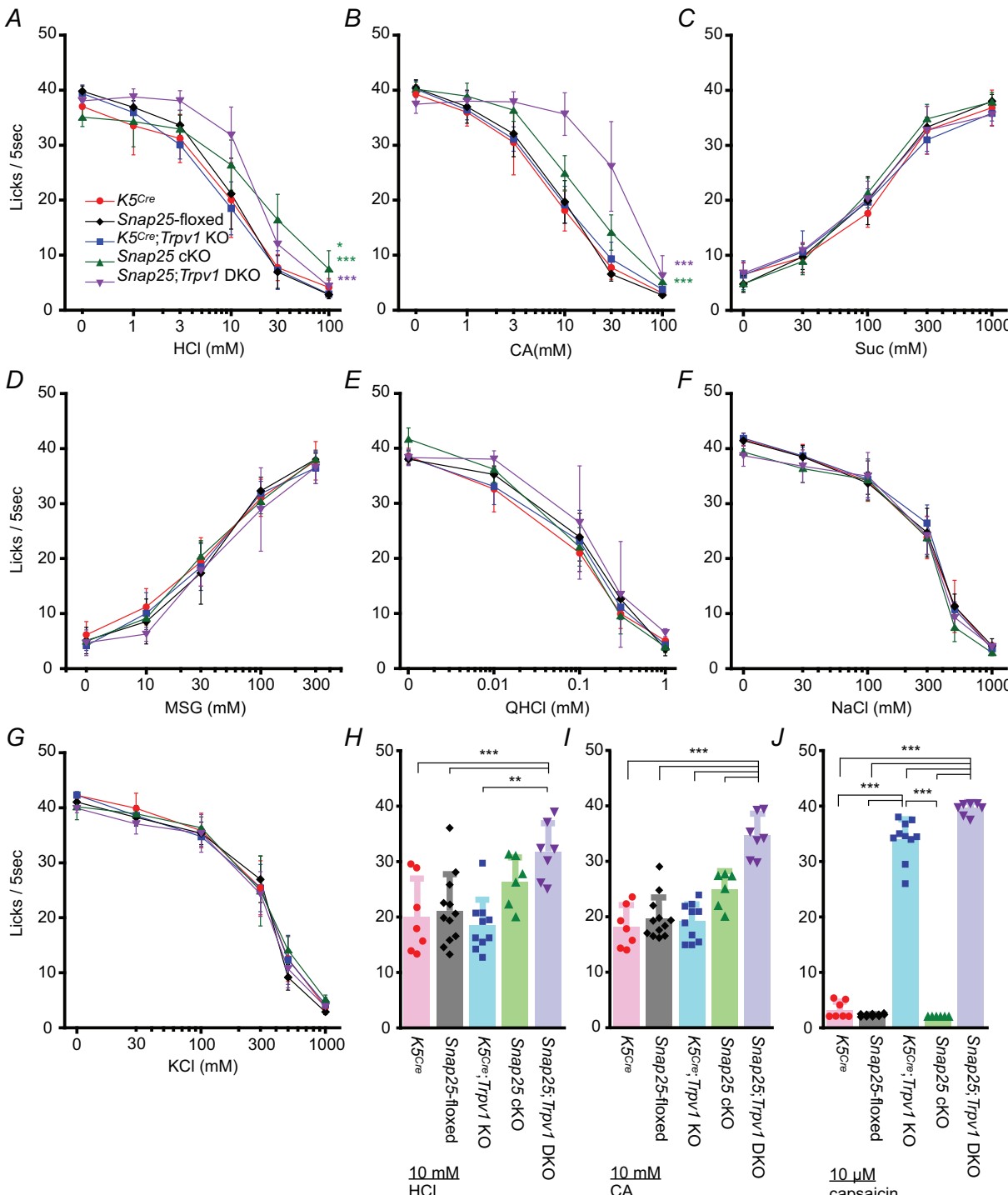

**Figure 6. Aversive responses to sour tastants were reduced in *Snap25* cKO mice**

*A–G*) short-term lick responses to hydrochloric acid (HCl) *A*), citric acid (CA) *B*), sucrose (Suc) *C*), monosodium glutamate (MSG) *D*), quinine-HCl (QHCl) *E*), NaCl *F*) and KCl *G*) in K5[Cre] (red), *Snap25*-floxed (black), K5[Cre];*Trpv1* KO (Blue), *Snap25* cKO (green) and *Snap25*;*Trpv1* DKO mice (purple). *H* and *I*) comparison of lick responses to 10 mM HCl *H*), 10 mM CA *I*) and 10 µM capsaicin among K5[Cre], *Snap25*-floxed, K5[Cre];*Trpv1* KO, *Snap25* cKO and *Snap25*;*Trpv1* DKO mice. \*\*\*$P < 0.001$ (*Snap25*;*Trpv1* DKO *vs.* K5[Cre], *Snap25*-floxed, K5[Cre];*Trpv1* KO in citric acid, and *Snap25*;*Trpv1* DKO *vs.* K5[Cre], K5[Cre];*Trpv1* KO in HCl), \*\*$P < 0.01$ (*Snap25* cKO *vs.* K5[Cre], *Snap25*-floxed in citric acid, and *Snap25* cKO *vs.* K5[Cre], *Snap25*;*Trpv1* DKO *vs.* *Snap25*-floxed in HCl), \*$P < 0.05$ (*Snap25* cKO *vs.* *Snap25*;*Trpv1* DKO, K5[Cre];*Trpv1* KO in citric acid, and *Snap25* cKO *vs.* K5[Cre];*Trpv1* KO in HCl). Two-way *A–G*) and one-way ANOVA *H–J*) with *post hoc* Tukey's HSD test. Animal, $n = 6$–12 per genotype. All data are presented as the mean ± SD. [Colour figure can be viewed at wileyonlinelibrary.com]

**Table 3. Summary of statistical analyses related to Fig. 6**

| Figure no. | Statistical method | | *P* value |
|---|---|---|---|
| 6*A* (HCl) | Two-way ANOVA | Genotype | < 0.001 |
| | | Concentration | < 0.001 |
| | | Genotype × Concentration interaction | < 0.001 |
| | Tukey's HSD | *K5^{Cre} vs. Snap25* cKO | 0.003 |
| | | *K5^{Cre} vs. Snap25;Trpv1 DKO* | < 0.001 |
| | | *K5^{Cre} vs. K5^{Cre}; Trpv1 KO* | 1.00 |
| | | *K5^{Cre} vs. Snap25-floxed* | 0.692 |
| | | *Snap25 cKO vs. Snap25;Trpv1 DKO* | 0.675 |
| | | *Snap25 cKO vs. K5^{Cre}; Trpv1 KO* | 0.020 |
| | | *Snap25 cKO vs. Snap25-floxed* | 0.223 |
| | | *Snap25;Trpv1 DKO vs. K5^{Cre}; Trpv1 KO* | < 0.001 |
| | | *Snap25;Trpv1 DKO vs. Snap25-floxed* | 0.004 |
| | | *K5Cre; Trpv1 KO vs. Snap25-floxed* | 0.632 |
| 6*B* (CA) | Two-way ANOVA | Genotype | < 0.001 |
| | | Concentration | < 0.001 |
| | | Genotype × Concentration interaction | < 0.001 |
| | Tukey's HSD | *K5^{Cre} vs. Snap25* cKO | 0.006 |
| | | *K5^{Cre} vs. Snap25;Trpv1 DKO* | < 0.001 |
| | | *K5^{Cre} vs. K5^{Cre}; Trpv1 KO* | 0.950 |
| | | *K5^{Cre} vs. Snap25-floxed* | 0.982 |
| | | *Snap25 cKO vs. Snap25;Trpv1 DKO* | 0.016 |
| | | *Snap25 cKO vs. K5^{Cre}; Trpv1 KO* | 0.018 |
| | | *Snap25 cKO vs. Snap25-floxed* | 0.008 |
| | | *Snap25;Trpv1 DKO vs. K5^{Cre}; Trpv1 KO* | < 0.001 |
| | | *Snap25;Trpv1 DKO vs. Snap25-floxed* | < 0.001 |
| | | *K5Cre; Trpv1 KO vs. Snap25-floxed* | 0.999 |
| 6*C* (Suc) | Two-way ANOVA | Genotype | 0.763 |
| | | Concentration | < 0.001 |
| | | Genotype × Concentration interaction | 0.03 |
| 6*D* (MSG) | Two-way ANOVA | Genotype | 0.499 |
| | | Concentration | < 0.001 |
| | | Genotype × Concentration interaction | 0.066 |
| 6*E* (QHCl) | Two-way ANOVA | Genotype | 0.104 |
| | | Concentration | < 0.001 |
| | | Genotype × Concentration interaction | 0.031 |
| 6*F* (NaCl) | Two-way ANOVA | Genotype | 0.200 |
| | | Concentration | < 0.001 |
| | | Genotype × Concentration interaction | 0.474 |
| 6*G* (KCl) | Two-way ANOVA | Genotype | 0.478 |
| | | Concentration | < 0.001 |
| | | Genotype × Concentration interaction | 0.076 |
| 6*H* (10 mm HCl) | One-way ANOVA | | < 0.001 |
| | Tukey's HSD | *K5^{Cre} vs. Snap25* cKO | 0.278 |
| | | *K5^{Cre} vs. Snap25;Trpv1 DKO* | 0.004 |
| | | *K5^{Cre} vs. K5^{Cre}; Trpv1 KO* | 0.982 |
| | | *K5^{Cre} vs. Snap25-floxed* | 0.993 |
| | | *Snap25 cKO vs. Snap25;Trpv1 DKO* | 0.446 |
| | | *Snap25 cKO vs. K5^{Cre}; Trpv1 KO* | 0.074 |
| | | *Snap25 cKO vs. Snap25-floxed* | 0.374 |
| | | *Snap25;Trpv1 DKO vs. K5^{Cre}; Trpv1 KO* | < 0.001 |
| | | *Snap25;Trpv1 DKO vs. Snap25-floxed* | 0.003 |
| | | *K5Cre; Trpv1 KO vs. Snap25-floxed* | 0.810 |

*(Continued)*

**Table 3. (Continued)**

| Figure no. | | Statistical method | *P* value |
|---|---|---|---|
| 6*I* (10 mм CA) | One-way ANOVA | | < 0.001 |
| | Tukey's HSD | *K5^Cre vs. Snap25* cKO | 0.016 |
| | | *K5^Cre vs. Snap25;Trpv1 DKO* | < 0.001 |
| | | *K5^Cre vs. K5^Cre; Trpv1 KO* | 0.978 |
| | | *K5^Cre vs. Snap25-floxed* | 0.901 |
| | | *Snap25 cKO vs. Snap25;Trpv1 DKO* | < 0.001 |
| | | *Snap25 cKO vs. K5^Cre; Trpv1 KO* | 0.033 |
| | | *Snap25 cKO vs. Snap25-floxed* | 0.051 |
| | | *Snap25;Trpv1 DKO vs. K5^Cre; Trpv1 KO* | < 0.001 |
| | | *Snap25;Trpv1 DKO vs. Snap25-floxed* | < 0.001 |
| | | *K5Cre; Trpv1 KO vs. Snap25-floxed* | 0.998 |
| 6*J* (10 μm capsaicin) | One-way ANOVA | | < 0.001 |
| | Tukey's HSD | *K5^Cre vs. Snap25* cKO | 0.819 |
| | | *K5^Cre vs. Snap25;Trpv1 DKO* | < 0.001 |
| | | *K5^Cre vs. K5^Cre; Trpv1 KO* | < 0.001 |
| | | *K5^Cre vs. Snap25-floxed* | 0.852 |
| | | *Snap25 cKO vs. Snap25;Trpv1 DKO* | < 0.001 |
| | | *Snap25 cKO vs. K5^Cre; Trpv1 KO* | < 0.001 |
| | | *Snap25 cKO vs. Snap25-floxed* | 0.999 |
| | | *Snap25;Trpv1 DKO vs. K5^Cre; Trpv1 KO* | < 0.001 |
| | | *Snap25;Trpv1 DKO vs. Snap25-floxed* | < 0.001 |
| | | *K5Cre; Trpv1 KO vs. Snap25-floxed* | < 0.001 |

cells to the gustatory nerve (Finger et al., 2005; Huang et al., 2008). Although a drastic reduction in CT nerve responses to $NH_4Cl$ was also observed in *Snap25* cKO mice, a small response above the baseline was still detected (Fig. 5*B*). This has also been observed in individual CT nerve responses from *Otop1* KO and *Otop1/Skn1a* DKO mice (Liang et al., 2023). We speculate that $NH_4Cl$ taste is detected not only by type III cells, but also through an unknown pathway in the taste buds, possibly involving bitter sensitive type II taste cells, as $NH_4Cl$ has some bitter taste characteristics. Type III cells also respond to high salt through an amiloride-insensitive pathway (Oka et al., 2013; Taruno et al., 2021; Yoshida et al., 2009). Our study observed no significant increase in high salt licking behaviour (Fig. 6*F* and *G*) and no decrease in CT nerve responses to high salt (Fig. 5*H*) in *Snap25* cKO mice. To alter behavioral responses to high salt, other pathways, such as those involving bitter type II cells, may need to be simultaneously ablated in *Snap25* cKO mice (Oka et al., 2013). Regarding CT nerve responses to high salt, reducing variability in high salt responses in CT nerve recordings may help to further understand the type III cell-dependent high salt pathway. CT nerve responses to 10–30 mм HCl were greater than those to the same concentration of CA in *Snap25*-floxed mice (Fig. 5*C* and *D*). A similar trend was observed in *Snap25;Trpv1* DKO mice, which exhibited stronger avoidance of HCl compared to CA (Fig. 6*A* and *B*). These differences may be explained by variations in the p$K_a$ values of the solutions. HCl has a higher p$K_a$ ($\sim$−6.3) compared to CA ($\sim$3.13), indicating that HCl produces more protons than CA at the same concentration. This difference may result in a stronger signal in type III cells and other sensory pathways, leading to increased CT nerve response and avoidance behaviour in mice.

The reduction in CT nerve responses in *Snap25* cKO mice suggests that synaptic vesicles may serve as a source of ATP required for sour taste transmission in type III cells. Previous studies have strongly indicated that ATP, potentially released from type III cells, plays a crucial role in activating gustatory nerves (Finger et al., 2005; Kinnamon & Finger, 2013). Notably, the absence of P2X receptors significantly reduces nerve responses to acid stimuli (Finger et al., 2005). In our study, we demonstrated that the loss of synaptic vesicular release leads to a reduction in nerve responses to acid stimuli (Fig. 5). Taken together, our findings may support the idea that ATP is released through vesicular synapses in type III cells, although further experimental validation is needed.

In line with the deficiency of sour taste licks in *Otop1 KO* mice with the ablation of *Trpv1*-expressing trigeminal neurons (Zhang et al., 2019), *Snap25;Trpv1* DKO mice exhibited higher lick responses to sour tastants compared to other types of mice (Fig. 6), suggesting that sour taste signals are transmitted through at least two different pathways: taste and somatosensory pathways. Previous

studies have also revealed that these two types of pathways are crucial for sour sensation (Leffler et al., 2006; Turner & Liman, 2022; Zhang et al., 2019). However, *Snap25;Trpv1* DKO mice still showed aversive responses to sour compounds, especially at high concentrations (Fig. 6*A* and *B*). This suggests the presence of a third mechanism involved in sour taste detection. One of the potential mechanisms may be TRPA1-mediated acid sensation, which may contribute to somatosensory sour sensation, similar to TRPV1 in the trigeminal nerve (Dhaka et al., 2009). Testing licking behaviour in *Snap25;Trpv1;Trpa1* triple KO mice may be a key to further elucidate the complex physiological mechanisms of sour signal transmission *in vivo* (Rhyu et al., 2021). Another possibility is that the superior larynx (SL) may participate in sour sensation in mice. Mice lacking taste nerve responses from the CT nerve still exhibit electrophysiological responses of the SL to sour solutions and normal sour detection in the taste behavioural test, which suggests that the CT and SL nerves independently contribute to sour sensation *in vivo* (Ohkuri et al., 2012). Another possible pathway is olfaction, which is also responsible for detecting acids in the nasal cavity (Mosienko et al., 2017). The trigeminal nerve in the nasal cavity expresses *Trpv1* and the olfactory epithelium contains sour-sensitive olfactory receptors (Pluznick et al., 2013; Saunders et al., 2013). In the present study, although *Snap25;Trpv1* DKO mice lack both oral and nasal TRPV1-mediated somatosensation, their olfactory receptors remain intact. Therefore, *Snap25* cKO mice may retain an intact sour sensation through the olfactory system, which may influence their remaining sour avoidance. Taken together, our study further highlights the redundancy of pathways in sour sensation, which serves as a self-defense mechanism to detect foods to be rejected.

In conclusion, SNAP25 is a critical protein that mediates sour taste transmission in type III cells in mice by controlling synaptic vesicular release and contributing to cell survival of these cells.

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

## Additional information

### Data availability statement

All raw data are available via figshare (https://figshare.com/s/f72be29fad1cef7dc02a).

### Competing interests

The authors declare that they have no competing interests.

### Author contributions

K.H. and R.Y. were responsible for conceptualization. All authors were responsible for methodology. K.H., K.W., H.H., Y.M. and R.Y. were responsible for investigations. K.H. and R.Y. were responsible for writing the original draft. All authors were responsible for reviewing and editing. R.Y. was responsible for funding acquisition. K.H. and R.Y. were responsible for resources. RY was responsible for supervision.

### Funding

This work was supported by Japan Society for the Promotion of Science KAKENHI grants 21H03106, 23K21484, 24K22186 (RY), Umami Manufacturers Association (RY) and the Food Science Institute Foundation (RY). The funding source had no role in the design of the study; in the collection, analysis and interpretation of data; or in writing the manuscript.

### Acknowledgements

We thank Dr Junji Takeda (Osaka University, Japan), Dr Makoto Tominaga (National Institute for Physiological Sciences, Japan) and Dr Lee Anna Cunningham (University of New Mexico, USA) for providing the original stock of K5Cre, Trpv1 KO and Sanp25-floxed mice, respectively.

### Keywords

sour taste, synapse, taste buds, taste nerve, Type III cells

## Supporting information

Additional supporting information can be found online in the Supporting Information section at the end of the HTML view of the article. Supporting information files available:

**Peer Review History**

