## [Peer Review History · The Journal of Physiology]

Dual functions of SNAP25 in mouse taste buds

Kengo Horie, Kuanyu Wang, Hai Huang, Keiko Yasumatsu, Yuzo Ninomiya, Yoshihiro Mitoh, and Ryusuke Yoshida
DOI: 10.1113/JP288683

Corresponding author(s): Ryusuke Yoshida (yoshida.ryusuke@okayama-u.ac.jp)

The following individual(s) involved in review of this submission have agreed to reveal their identity: Sue C Kinnamon (Referee #1)

Review Timeline:

Submission Date:	03-Feb-2025
Editorial Decision:	05-Mar-2025
Revision Received:	31-Mar-2025
Editorial Decision:	17-Apr-2025
Revision Received:	21-Apr-2025
Editorial Decision:	25-Apr-2025
Revision Received:	01-May-2025
Accepted:	06-May-2025

Senior Editor: Nathan Schoppa

Reviewing Editor: Nathan Schoppa

Transaction Report:

Dear Dr Yoshida,

Re: JP-RP-2025-288683 "Dual functions of SNAP25 in mouse taste buds" by Kengo Horie, Kuanyu Wang, Hai Huang, Keiko Yasumatsu, Yuzo Ninomiya, Yoshihiro Mitoh, and Ryusuke Yoshida

Thank you for submitting your manuscript to The Journal of Physiology. It has been assessed by a Reviewing Editor and by 2 expert referees and we are pleased to tell you that it is acceptable for publication following satisfactory revision.

REVISION CHECKLIST:

We look forward to receiving your revised submission.

Yours sincerely,

Nathan Schoppa
Senior Editor
The Journal of Physiology

REQUIRED ITEMS

- The contact information for the person responsible for 'Research Governance' at your institution needs to be provided. This includes their name and an institutional email address. Please ensure the contact is not an author on this paper and provide an alternate contact if necessary, or confirm in the submission form that the author whose email was provided has sole responsibility for research governance. This is the person who is responsible for regulations, principles and standards of good practice in research carried out at the institution, for instance the ethical treatment of animals, the keeping of proper experimental records or the reporting of results.

- The reference list must be in alphabetical order, rather than numbered, to comply with our Journal format.

- Your manuscript must include a complete Additional Information section, including competing interests; funding; author contributions and acknowledgements.

- The Journal of Physiology funds authors of provisionally accepted papers to use the premium BioRender site to create high resolution schematic figures. Follow this link and enter your details and the manuscript number to create and download figures. Upload these as the figure files for your revised submission. If you choose not to take up this offer, we require figures to be of similar quality and resolution. If you are opting out of this service to authors, state this in the Comments section on the Detailed Information page of the submission form. The link provided should only be used for the purposes of this submission. Authors will be charged for figures created on this premium BioRender account if they are not related to this manuscript submission.

- Please upload separate high-quality figure files via the submission form.

- A Data Availability Statement is required for all papers reporting original data. This must be in the Additional Information section of the manuscript itself. It must have the paragraph heading 'Data Availability Statement'. All data supporting the results in the paper must be either: in the paper itself; uploaded as Supporting Information for Online Publication; or archived in an appropriate public repository. The statement needs to describe the availability or the absence of shared data. Authors must include in their statement: a link to the repository they have used, or a statement that it is available as Supporting Information; reference the data in the appropriate sections(s) of their manuscript; and cite the data they have shared in the References section. Whenever possible, the scripts and other artefacts used to generate the analyses presented in the paper should also be publicly archived. If sharing data compromises ethical standards or legal requirements then authors are not expected to share it, but must note this in their statement. For more information, see our Statistics Policy.

- The contact information for the person responsible for 'Research Governance' at your institution needs to be provided. This includes their name and an institutional email address. Please ensure the contact is not an author on this paper and provide an alternate contact if necessary, or confirm in the submission form that the author whose email was provided has sole responsibility for research governance. This is the person who is responsible for regulations, principles and standards of good practice in research carried out at the institution, for instance the ethical treatment of animals, the keeping of proper experimental records or the reporting of results.

- The reference list must be in alphabetical order, rather than numbered, to comply with our Journal format.
- Your manuscript must include a complete Additional Information section, including competing interests; funding; author contributions and acknowledgements.
- The Journal of Physiology funds authors of provisionally accepted papers to use the premium BioRender site to create high resolution schematic figures. Follow this link and enter your details and the manuscript number to create and download figures. Upload these as the figure files for your revised submission. If you choose not to take up this offer, we require figures to be of similar quality and resolution. If you are opting out of this service to authors, state this in the Comments section on the Detailed Information page of the submission form. The link provided should only be used for the purposes of this submission. Authors will be charged for figures created on this premium BioRender account if they are not related to this manuscript submission.
- Please upload separate high-quality figure files via the submission form.
- Papers must comply with the Statistics Policy: https://jp.msubmit.net/cgi-bin/main.plex?form_type=display_requirements#statistics.

In summary:

- If $n \leq 30$, all data points must be plotted in the figure in a way that reveals their range and distribution. A bar graph with data points overlaid, a box and whisker plot or a violin plot (preferably with data points included) are acceptable formats.
- If $n > 30$, then the entire raw dataset must be made available either as supporting information, or hosted on a not-for-profit repository, e.g. FigShare, with access details provided in the manuscript.
- 'n' clearly defined (e.g. x cells from y slices in z animals) in the Methods. Authors should be mindful of pseudoreplication.
- All relevant 'n' values must be clearly stated in the main text, figures and tables.
- The most appropriate summary statistic (e.g. mean or median and standard deviation) must be used. Standard Error of the Mean (SEM) alone is not permitted.
- Exact p values must be stated. Authors must not use 'greater than' or 'less than'. Exact p values must be stated to three significant figures even when 'no statistical significance' is claimed.
- A Data Availability Statement is required for all papers reporting original data. This must be in the Additional Information section of the manuscript itself. It must have the paragraph heading 'Data Availability Statement'. All data supporting the results in the paper must be either: in the paper itself; uploaded as Supporting Information for Online Publication; or archived in an appropriate public repository. The statement needs to describe the availability or the absence of shared data. Authors must include in their statement: a link to the repository they have used, or a statement that it is available as Supporting Information; reference the data in the appropriate sections(s) of their manuscript; and cite the data they have shared in the References section. Whenever possible, the scripts and other artefacts used to generate the analyses presented in the paper should also be publicly archived. If sharing data compromises ethical standards or legal requirements then authors are not expected to share it, but must note this in their statement. For more information, see our Statistics Policy.

EDITOR COMMENTS

Senior Editor:

Comments for Authors to ensure the paper complies with the Statistics Policy:
Precise p-values need to be provided unless $p < 0.001$.

Comments to the Author:

Your manuscript has been reviewed by two expert referees, who both felt that the study was well done and convincing. Also, your results showing that loss of SNAP25 -- and by implication loss of vesicular exocytosis -- impacted both information flow and maintenance of Type III taste cells was considered to be quite novel and impactful. The referees only had essentially minor concerns about the presentation and discussion of results, all of which will need to be addressed. I also have a couple of concerns. The following points summarize most of the concerns:

1. The authors should clarify the background and significance of the results, following comments from both reviewers.
2. The authors should improve the presentation of some of the figure panels.
3. Referee 2 requested further explanation about some of the results.
4. Journal of Physiology does not allow supporting figures in final publications. Results that are in supporting figures should be included as panels in main figures or described in the text.
5. Precise p-values need to be provided when $p < 0.001$.

Referee 1 suggests reordering the latter figures in the study and the associated text. This change is not required, but the authors should consider whether the change would improve the presentation.

REFEREE COMMENTS

Referee #1:

General Comments:

The manuscript by Wang et al. addresses an important problem in taste physiology, the role of synaptic vesicles in the transmission of taste information to afferent nerve fibers. Type III cells, which detect and transduce sour (acidic) stimuli, are the only taste cell type that makes conventional, vesicular synapses with afferent nerve fibers. While it is known that sour taste stimuli elicit release of 5-HT from Type III cells, likely through vesicular release (and VMATs), 5-HT is only partially responsible for the afferent nerve response to sour stimuli, whereas ATP acting on P2X neural receptors is crucial to the response. Yet Type III cells do not express vesicular ATP transporters (VNUTs) leading to the question as to the exact role synaptic vesicular release may play in the activation of nerve fibers by Type III cells. Some of these ideas should be made more explicit in the introduction and/or discussion.

To address this question, the authors generated an epithelial-specific conditional knockout of SNAP25 -- a snare protein essential for vesicular synaptic transmission -- and assessed the effect of the deletion of SNAP25 on the peripheral taste system including assessment of taste cell types using taste cell-specific immunocytochemical markers and electrophysiological and behavioral response to acidic stimuli. Nerve and behavioral responses to acids were markedly reduced. Remarkably, deletion of SNAP25 resulted in a significant reduction in the number of Type III taste cells compared to floxed control mice. To determine if the reduction in the number of Type III cells was due to a lack of taste cell proliferation or to a failure to maintain most-mitotic Type III cells, the authors injected EdU into both conditional knockout and floxed control mice and sacrificed them at either 7 or 14 days post injection. After 7 days post EdU injection, there was no effect of SNAP25 deletion on the number of Type III cells, suggesting that Type III cell proliferation and differentiation was not impaired by the deletion. After 14 days, however, there were significant decreases in Type III cell numbers compared to control mice, while the numbers of other taste cell types were unchanged. These data imply that long-term maintenance of Type III cells is specifically impacted by the lack of vesicular exocytosis, while other cell types that do not use vesicular transmission are spared and have a normal life span. These data are compelling and novel and suggest that vesicular exocytosis is required for the survival and maintenance of Type III taste cells. In general, the experiments in this manuscript are well designed, rigorous, and adequately controlled, and should be of considerable interest to the scientific community. I have minor specific suggestions for improvement, listed below.

One suggestion is that the ordering of the results in the paper be changed to present the nerve data prior to the behavioral data so that the results flow from cellular level to neural level to behavior. Conceptually, this seems more clear that jumping from behavior back down to neural responses.

Specific Suggestions:

1. P. 5, line 132: hydrogen chloride should be hydrochloric acid
2. P. 5, line 139: eliminate "with"
3. P 9 Fig. 1 caption: the word "slice" should be replaced by "section".
4. Figures: All image figures of single taste buds (e.g. 1A, 2A-D) need higher magnification for results to be easily visualized. This can be accomplished easily with the existing figures simply by cropping away the large areas of black space surrounding the buds and enlarging the resulting image to occupy the same space on the page.
5. Fig. 1B: An important point is that the gustatory nerve fibers retain SNAP25 immunoreactivity, thereby reinforcing the idea that the KO is cell type specific and not likely to affect neural transmission. This can be seen in Fig. 1B only by making the center column of figures (SNAP channel) brighter. This can be accomplished by changing the gamma of the red-blue channels. This sort of adjustment is not necessary to visualize stained cells, as in Fig 2A-D, but is crucial to visualize the nerve fibers in Fig. 1.
6. P. 13, line 231. The use of the term "proliferating" is incorrect. These cells are post-mitotic and not proliferative. Please use the term "post mitotic" when describing this population of cells. To distinguish between the 7- and 14- day results, the authors might use the phrases: "younger Type III cells (< 8 days)" and "older Type 3 cells (> 8 days)".
7. P. 19, line 273. Induce should be "eliminate"
8. P. 23, lines 341 and 345. Taste cell degeneration was not specifically observed in these studies. The degeneration is assumed because of decreased cell numbers, but is not demonstrated directly herein. "Elimination" or "loss" would be a better term for this phenomenon.
9. P. 24, line 355. Taste and somatosensory pathways have been suggested numerous times before for sour transduction. The previous studies should be referenced here. But the authors are not really suggesting here that another modality is involved in avoidance. They are instead suggesting that TrpA1 plays an additional role in somatosensory -mediated aversion.
10. Discussion: I think the authors could go further and speculate that their experiments suggest that synaptic vesicles in Type III cells may be the source of the ATP that is required for sour taste transmission. ATP release has not been measured previously from Type III taste cells, but if sour taste transmission is completely blocked by blocking vesicular exocytosis, this suggests that the ATP must be coming from the vesicles that also contain 5-HT and other transmitters. The source of ATP for Type III transmission has been a long unsolved question in the taste field

Referee #2:

The manuscript by Horie et al describes an innovative and important approach to investigating taste buds, specifically sour-sensing (Type III) taste receptor cells. The work is carried out well and documented with care. Essentially, Horie et al show that Type III taste bud cells depend on SNAP25 expression for maturation and function. In the absence of SNAP25, Type III (sour-sensing) taste buds receptor cells fail to develop. The authors used an innovative way to eliminate SNAP25 in taste bud cells without affecting global expression of this synaptic protein throughout the nervous system. The researchers use tissue immunostaining, taste behavioral assays, and whole nerve recordings of the chorda tympani nerve to analyze taste function in mice which lack SNAP25 expression in taste buds. In addition to the experiments being carefully carried out, the manuscript has been written quite well. The report is quite readable. The results will likely have a significant impact in the field of sensory neurobiology and will provide insight into synaptic function of Type III taste bud cells.

There are no significant flaws in the experimentation or documentation. Appropriate controls (SNAP25 floxed mice) are included in all experiments. Differences between SNAP25 conditional KO mice and controls are striking. The conclusions are convincing.

There are a few instances, however, where the authors may wish to clarify or explain certain points to strengthen their manuscript, as follows:

Line 64: what is a "*channel synapse*"? This is not conventional terminology. Presumably the authors mean that Type II taste cells release transmitter via a novel synaptic channel, CALHM1/3.

Line 66 typo: "CHALM1/3" should be CALHM1/3

Fig. 1A,B It would be helpful to have discrete, dotted lines outlining the taste buds. It's not clear a non-specialist will recognize the structures as taste buds. Ideally, the micrographs should be combined with differential contrast optics or some way to visualize the tissue and show the taste buds in context. But failing this, dotted/dashed lines might help. Same applies to Fig S2.

Legends to figures (e.g. line 198) should include number of animals as well as number of taste buds included in the samples. Some of the legends have this information, some do not.

Fig. 2F: The spread (variation) of data points for SNAP25 cK/O, 7 days, is remarkably narrow compared with the spread of data points for the control tissue (SNAP25 floxed, 7 day). This seems unusual. Is this a result of limited sample size? Or is there some biological significance in this difference?

Fig. 3A,B. SNAP25,TRPV1 DKO mice seem to have less of an avoidance to citric acid than to HCl in the 10-30 mM range. Do the authors have some explanation for this? Interestingly, a similar differential also can be seen in the whole nerve recordings (Fig 4 C,D).

Fig. 4 legend: it would be helpful if the acronyms for the different acids were spelled out in the legend.

Line 334: the authors may wish to be more circumspect about stating "...we are the first to demonstrate...". Although strictly speaking this statement may be true, most research publications are the "first" to demonstrate a particular finding. Such a statement as "we are the first" generates a sense that the authors seem boastful and self-promoting. A more neutral statement such as "Our findings highlight the presence of these two distinct signal transmission mechanisms in the peripheral taste system and demonstrate that vesicular synaptic transmission in taste buds is essential..." would be more professional.

Line 355-358: An important factor that likely contributes to an aversion to HCl and citric acid solutions is **olfaction**. It is highly likely that the odor of HCl and citric acid is aversive and reduces the desire to lick these solutions. Olfaction in SNAP25;TRPV1 DKO mice is presumably still relatively intact (though this was not tested) and this may contribute to why these mice still avoid acid solutions.

END OF COMMENTS

Dear Editor,

Thank you for reviewing our paper and for your thoughtful comments, which have significantly improved its quality. We have carefully addressed each comment and made the necessary revisions to the manuscript. The responses to the Editor, Referees 1, and 2 have been highlighted in magenta, blue, and yellow, respectively, within the manuscript. All our responses are written in Times New Roman.

EDITOR COMMENTS

Senior Editor:

Comments for Authors to ensure the paper complies with the Statistics Policy:

Precise p-values need to be provided unless $p < 0.001$.

Comments to the Author:

Your manuscript has been reviewed by two expert referees, who both felt that the study was well done and convincing. Also, your results showing that loss of SNAP25 -- and by implication loss of vesicular exocytosis -- impacted both information flow and maintenance of Type III taste cells was considered to be quite novel and impactful. The referees only had essentially minor concerns about the presentation and discussion of results, all of which will need to be addressed. I also have a couple of concerns. The following points summarize most of the concerns:

1. The authors should clarify the background and significance of the results, following comments from both reviewers.

We have revised the manuscript in accordance with the Referee's comments. All details are provided in our responses to each comment.

2. The authors should improve the presentation of some of the figure panels.

We have revised the figures and adjusted the brightness and contrast in accordance with the Referee's comments. Please refer to our responses to the Referee's comments.

3. Referee 2 requested further explanation about some of the results.

We have added explanations to the manuscript. Please refer to our responses to Referee 2's comments for further details.

4. Journal of Physiology does not allow supporting figures in final publications. Results that are in supporting figures should be included as panels in main figures or described in the text.

Former Supplementary Figures 1 and 2 have been incorporated into Figures 1 and 2. Likewise, former Supplementary Figures 3 and 4 have been merged into Figures 3 and 4. Please refer to our response below for further details.

5. Precise p-values need to be provided when $p < 0.001$.

We have provided precise p-values throughout the manuscript when they were $p > 0.001$. During our careful verification of each p-value, we found that a two-way ANOVA revealed an interaction effect of Treatment day \times Genotype in the EdU-CA4 tracing experiments in Figure 3B (previously Figure 2) with $p = 0.0770$, indicating a trend toward interaction but not statistical significance. The main effects of Treatment day ($p < 0.001$) and Genotype ($p = 0.0162$) were significant, suggesting an overall genotype effect in reducing the number of EdU-positive CA4 taste cells in Snap25 cKO animals. Although this represents a trend toward significance, the graphical representation motivated us to further conduct a post hoc analysis as an exploratory experiment. In addition to these findings, an analysis from another perspective further supports the significance of this phenotype. Although this finding does not alter our overall conclusion, we have rephrased our statements to clarify our findings. The relevant sentences in the Abstract, Results, and Discussion sections have been carefully revised and highlighted in the manuscript.

Referee 1 suggests reordering the latter figures in the study and the associated text. This change is not required, but the authors should consider whether the change would improve the presentation.

We have revised the figures in accordance with Referee 1's suggestion by incorporating the supplementary figures into the main figures and reordering them as follows:

- Former Supplementary Figures 1 and 2 have been integrated into Figures 1 and 2, respectively, and their panels have been reordered.
- Former Figure 2 has been divided and renamed as Figures 3 and 4, with panel reordering.
- Former Supplementary Figures 3 and 4 have been merged into Figures 3 and 4, respectively.
- Former Figures 4 and 3 have been renamed and reordered as Figures 5 and 6, respectively.

REFeree COMMENTS

Referee #1:

General Comments:

The manuscript by Wang et al. addresses an important problem in taste physiology, the role of synaptic vesicles in the transmission of taste information to afferent nerve fibers. Type III cells, which detect and transduce sour (acidic) stimuli, are the only taste cell type that makes conventional, vesicular synapses with afferent nerve fibers. While it is known that sour taste stimuli elicit release of 5-HT from Type III cells, likely through vesicular release (and VMATs), 5-HT is only partially responsible for the afferent nerve response to sour stimuli, whereas ATP acting on P2X neural receptors is crucial to the response. Yet Type III cells do not express vesicular ATP transporters (VNUTs) leading to the question as to the exact role synaptic vesicular release may play in the activation of nerve fibers by Type III cells. Some of these ideas should be made more explicit in the introduction and/or discussion.

Thank you for highlighting the key background information on Type III cells. We have incorporated sentences introducing these points into the Introduction section.

To address this question, the authors generated an epithelial-specific conditional knockout of SNAP25 -- a snare protein essential for vesicular synaptic transmission -- and assessed the effect of the deletion of SNAP25 on the peripheral taste system including assessment of taste cell types using taste cell-specific immunocytochemical markers and electrophysiological and behavioral response to acidic stimuli. Nerve and behavioral responses to acids were markedly reduced. Remarkably, deletion of SNAP25 resulted in a

significant reduction in the number of Type III taste cells compared to floxed control mice. To determine if the reduction in the number of Type III cells was due to a lack of taste cell proliferation or to a failure to maintain most-mitotic Type III cells, the authors injected EdU into both conditional knockout and floxed control mice and sacrificed them at either 7 or 14 days post injection. After 7 days post EdU injection, there was no effect of SNAP25 deletion on the number of Type III cells, suggesting that Type III cell proliferation and differentiation was not impaired by the deletion. After 14 days, however, there were significant decreases in Type III cell numbers compared to control mice, while the numbers of other taste cell types were unchanged. These data imply that long-term maintenance of Type III cells is specifically impacted by the lack of vesicular exocytosis, while other cell types that do not use vesicular transmission are spared and have a normal life span. These data are compelling and novel and suggest that vesicular exocytosis is required for the survival and maintenance of Type III taste cells. In general, the experiments in this manuscript are well designed, rigorous, and adequately controlled, and should be of considerable interest to the scientific community. I have minor specific suggestions for improvement, listed below.

One suggestion is that the ordering of the results in the paper be changed to present the nerve data prior to the behavioral data so that the results flow from cellular level to neural level to behavior. Conceptually, this seems more clear than jumping from behavior back down to neural responses.

Thank you for suggesting the reordering of figures. We have revised the figure order in accordance with your comment. Please refer to our response to Editor's comment 4 for further details.

Specific Suggestions:

1. P. 5, line 132: hydrogen chloride should be hydrochloric acid

We have corrected it.

2. P. 5, line 139: eliminate "with"

We have removed it.

3. P 9 Fig. 1 caption: the word "slice" should be replaced by "section".

We have corrected it.

4. Figures: All image figures of single taste buds (e.g. 1A, 2A-D) need higher magnification for results to be easily visualized. This can be accomplished easily with the existing figures simply by cropping away the large areas of black space surrounding the buds and enlarging the resulting image to occupy the same space on the page.

We have magnified those figures.

5. Fig. 1B: An important point is that the gustatory nerve fibers retain SNAP25 immunoreactivity, thereby reinforcing the idea that the KO is cell type specific and not likely to affect neural transmission. This can be seen in Fig. 1B only by making the center column of figures (SNAP channel) brighter. This can be accomplished by changing the gamma of the red-blue channels. This sort of adjustment is not necessary to visualize stained cells, as in Fig 2A-D, but is crucial to visualize the nerve fibers in Fig. 1.

We have adjusted the brightness and contrast of SNAP25 and Merge images in Figures 1–4 to enhance the immunoreactivity of nerve SNAP25 and improve the bright-field view.

6. P. 13, line 231. The use of the term "proliferating" is incorrect. These cells are post-mitotic and not proliferative. Please use the term "post mitotic" when describing this population of cells. To distinguish between the 7- and 14- day results, the authors might use the phrases: "younger Type III cells (< 8 days)" and "older Type 3 cells (> 8 days)".

We have corrected "proliferating" to "post-mitotic". Following a careful verification of our statistical data, we have revised our phrasing to more precisely describe the phenotype of *Snap25* cKO in this experiment. However, the overall conclusion remains unchanged. Please also refer to our response to Editor's comment 5 for further details.

7. P. 19, line 273. Induce should be "eliminate"

We have corrected it.

8. P. 23, lines 341 and 345. Taste cell degeneration was not specifically observed in these studies. The degeneration is assumed because of decreased cell numbers, but is not

demonstrated directly herein. "Elimination" or "loss" would be a better term for this phenomenon.

We have corrected them.

9. P. 24, line 355. Taste and somatosensory pathways have been suggested numerous times before for sour transduction. The previous studies should be referenced here. But the authors are not really suggesting here that another modality is involved in avoidance. They are instead suggesting that TrpA1 plays an additional role in somatosensory -mediated aversion.

We have added the sentence describing the previous studies.

10. Discussion: I think the authors could go further and speculate that their experiments suggest that synaptic vesicles in Type III cells may be the source of the ATP that is required for sour taste transmission. ATP release has not been measured previously from Type III taste cells, but if sour taste transmission is completely blocked by blocking vesicular exocytosis, this suggests that the ATP must be coming from the vesicles that also contain 5-HT and other transmitters. The source of ATP for Type III transmission has been a long unsolved question in the taste field

Thank you for your insightful suggestion. We have incorporated the proposed points into the Discussion section.

Referee #2:

The manuscript by Horie et al describes an innovative and important approach to investigating taste buds, specifically sour-sensing (Type III) taste receptor cells. The work is carried out well and documented with care. Essentially, Horie et al show that Type III taste bud cells depend on SNAP25 expression for maturation and function. In the absence of SNAP25, Type III (sour-sensing) taste buds receptor cells fail to develop. The authors used an innovative way to eliminate SNAP25 in taste bud cells without affecting global expression of this synaptic protein throughout the nervous system. The researchers use tissue immunostaining, taste behavioral assays, and whole nerve recordings of the chorda tympani nerve to analyze taste function in mice which lack SNAP25 expression in taste buds. In addition to the experiments being carefully carried out, the manuscript has been written quite well. The report is quite readable. The results will likely have a significant impact in the field of sensory neurobiology and will provide insight into synaptic function of Type III taste bud cells.

There are no significant flaws in the experimentation or documentation. Appropriate controls (SNAP25 floxed mice) are included in all experiments. Differences between SNAP25 conditional KO mice and controls are striking. The conclusions are convincing.

There are a few instances, however, where the authors may wish to clarify or explain certain points to strengthen their manuscript, as follows:

Line 64: what is a "*channel synapse*"? This is not conventional terminology. Presumably the authors mean that Type II taste cells release transmitter via a novel synaptic channel, CALHM1/3.

We have added the description of channel synapse.

Line 66 typo: "CHALM1/3" should be CALHM1/3

We have corrected it.

Fig. 1A,B It would be helpful to have discrete, dotted lines outlining the taste buds. It's not clear a non-specialist will recognize the structures as taste buds. Ideally, the micrographs should be combined with differential contrast optics or some way to visualize the tissue and show the taste buds in context. But failing this, dotted/dashed lines might help. Same applies to Fig S2.

We have added and enhanced the bright-field view of Merge images in Figures 1–4 to better visualize the morphology of taste buds.

Legends to figures (e.g. line 198) should include number of animals as well as number of taste buds included in the samples. Some of the legends have this information, some do not.

We have added the number of animals to the legends.

Fig. 2F: The spread (variation) of data points for SNAP25 cKO, 7 days, is remarkably narrow compared with the spread of data points for the control tissue (SNAP25 floxed, 7 day). This seems unusual. Is this a result of limited sample size? Or is there some biological significance in this difference?

We considered that this was due to the limited sample size. In contrast to *Snap25* cKO at Day 7, *Snap25*-floxed at Day 7 exhibited relatively large variation, suggesting greater variability within the population of each group. We have mentioned this point in the Result and Discussion sections.

Fig. 3A,B. SNAP25,TRPV1 DKO mice seem to have less of an avoidance to citric acid than to HCl in the 10-30 mM range. Do the authors have some explanation for this? Interestingly,

a similar differential also can be seen in the whole nerve recordings (Fig 4 C,D).

This may be explained by differences in the pKa values of the solutions. HCl has a higher pKa (~6.3) compared to CA (~3.13), indicating that HCl produces more protons than CA at the same concentration. This difference may result in stronger CT nerve responses and an enhanced avoidance reaction to HCl in mice. We have addressed this point in the Discussion section.

Fig. 4 legend: it would be helpful if the acronyms for the different acids were spelled out in the legend.

We have added them to the legends.

Line 334: the authors may wish to be more circumspect about stating "...we are the first to demonstrate...". Although strictly speaking this statement may be true, most research publications are the "first" to demonstrate a particular finding. Such a statement as "we are the first" generates a sense that the authors seem boastful and self-promoting. A more neutral statement such as "Our findings highlight the presence of these two distinct signal transmission mechanisms in the peripheral taste system and demonstrate that vesicular synaptic transmission in taste buds is essential..." would be more professional.

Thank you for highlighting this issue. We have removed the sentence.

Line 355-358: An important factor that likely contributes to an aversion to HCl and citric acid solutions is **olfaction**. It is highly likely that the odor of HCl and citric acid is aversive and reduces the desire to lick these solutions. Olfaction in SNAP25;TRPV1 DKO mice is presumably still relatively intact (though this was not tested) and this may contribute to why these mice still avoid acid solutions.

Thank you for pointing out another discussion. We have added the discussion for olfactory system.

Dear Dr Yoshida,

Re: JP-RP-2025-288683R1 "Dual functions of SNAP25 in mouse taste buds" by Kengo Horie, Kuanyu Wang, Hai Huang, Keiko Yasumatsu, Yuzo Ninomiya, Yoshihiro Mitoh, and Ryusuke Yoshida

Thank you for submitting your manuscript to The Journal of Physiology. It has been assessed by a Reviewing Editor and by 2 expert referees and we are pleased to tell you that it is acceptable for publication following satisfactory revision.

REVISION CHECKLIST:

We look forward to receiving your revised submission.

Yours sincerely,

Nathan Schoppa
Senior Editor
The Journal of Physiology

EDITOR COMMENTS

Thank you for the revised manuscript. The reviewers were generally pleased with the changes that were made and are quite positive about the overall impact of the study. There are still a few remaining minor concerns to be addressed around the reporting of statistics, including one point raised by reviewer 2. Reviewer 2 also raised a minor concern about the micrographs in Figure 1 that the authors should also consider before they send in their next revision.

The remaining concerns around statistics that should be addressed include:

1. For cases in which many statistical comparisons are made, the authors should summarize p-values in a table. This is most true for the results described around Figures 5 and 6. Using a table will make the results much more easily digestible for readers.
2. The authors should not discuss "trends" in their results, as they do around line 255. If $p > 0.05$, the authors should just indicate that there was not a significant effect, although they may wish to discuss the variabilities in results. Based on the authors' comments in the rebuttal letter in response to prior concerns raised about statistics (see editor's prior point 5), it appears that the lack of a significant effect around this particular result does not impact their conclusions. However, the authors should also re-word text elsewhere in the manuscript, if any re-wording is required.

REFEREE COMMENTS

Referee #1:

The authors have revised the manuscript and have answered all my questions adequately.

Referee #2:

The authors have responded positively and appropriately to all my comments. However, they may want to check their settings on the micrographs in Fig 1. The bright field or interference contrast fields (black and white) in the merged images seem very faint to me. It is still difficult to visualize the taste buds, and there still is the concern that many readers (non specialists) won't recognize the structures as taste buds. This is not a fatal flaw, however, just a suggestion. If the authors are satisfied with their micrographs, they can ignore this comment.

Similarly, the authors have responded to concerns about statistics by crowding their Results with multiple lines of details about effect size, etc. (e.g., lines 250ff). Horie et al are to be applauded for their attention to detail. However, all these lines of detail now make these paragraphs somewhat unreadable, or certainly bogs down the normal reader. These details might better fit into neat tables rather than embedded within the text. Again, this is the authors' decision to make and only a suggestion to make their nice report more readable.

END OF COMMENTS

Thank you for your comments regarding the statistical analyses and overall readability of our manuscript. We have carefully revised the manuscript and appropriately modified both the text and figures. All changes in the manuscript are highlighted in yellow.

Responses to the Editor's and Reviewers' comments are provided in Times New Roman font in this letter.

EDITOR COMMENTS

1. For cases in which many statistical comparisons are made, the authors should summarize p-values in a table. This is most true for the results described around Figures 5 and 6. Using a table will make the results much more easily digestible for readers.

Thank you for your comment. In response, we have created three tables summarizing the ANOVA analyses to improve the readability of the results.

2. The authors should not discuss "trends" in their results, as they do around line 255. If $p > 0.05$, the authors should just indicate that there was not a significant effect, although they may wish to discuss the variabilities in results. Based on the authors' comments in the rebuttal letter in response to prior concerns raised about statistics (see editor's prior point 5), it appears that the lack of a significant effect around this particular result does not impact their conclusions. However, the authors should also re-word text elsewhere in the manuscript, if any re-wording is required.

Thank you for pointing this out. After careful consideration, we concluded that this discussion was not necessary. Accordingly, we have removed the description of variability around line 255, as well as the exploratory post hoc analysis and significance bars from Figure 4B.

REFEREE COMMENTS

Referee #2:

The authors have responded positively and appropriately to all my comments. However, they may want to check their settings on the micrographs in Fig 1. The bright field or interference contrast fields (black and white) in the merged images seem very faint to me. It is still difficult to visualize the taste buds, and there still is the concern that many readers (non specialists) won't recognize the structures as taste buds. This is not a fatal flaw, however, just a suggestion. If the authors are satisfied with their micrographs, they can ignore this comment.

Thank you for your comment. To improve the visibility of the taste buds, we have outlined them with dotted lines in Figures 1 and 3.

Similarly, the authors have responded to concerns about statistics by crowding their Results with multiple lines of details about effect size, etc. (e.g., lines 250ff). Horie et al are to be applauded for their attention to detail. However, all these lines of detail now make these paragraphs somewhat unreadable, or certainly bogs down the normal reader. These details might better fit into neat tables rather than embedded within the text. Again, this is the authors' decision to make and only a suggestion to make their nice report more readable.

Thank you for your constructive comment. In response, we have prepared three tables that summarize the ANOVA results.

Dear Dr Yoshida,

Re: JP-RP-2025-288683R2 "Dual functions of SNAP25 in mouse taste buds" by Kengo Horie, Kuanyu Wang, Hai Huang, Keiko Yasumatsu, Yuzo Ninomiya, Yoshihiro Mitoh, and Ryusuke Yoshida

Thank you for submitting your manuscript to The Journal of Physiology. It has been assessed by a Reviewing Editor and by 0 expert referees and we are pleased to tell you that it is acceptable for publication following satisfactory revision.

REVISION CHECKLIST:

We look forward to receiving your revised submission.

Yours sincerely,

Nathan Schoppa
Senior Editor
The Journal of Physiology

REQUIRED ITEMS

- You must upload original, uncropped western blot/gel images (including controls) if they are not included in the manuscript. This is to confirm that no inappropriate, unethical or misleading image manipulation has occurred. These should be uploaded as 'Supporting information for review process only'. Please label/highlight the original gels so that we can clearly see which sections/lanes have been used in the manuscript figures. For more information, see: <https://physoc.onlinelibrary.wiley.com/hub/journal-policies#imagmanip>.

EDITOR COMMENTS

Senior Editor:

Thank you for the resubmission. We appreciate the new tables that summarize the statistics for many of the results. However, the authors still have not adequately addressed one of the concerns that I previously raised. They still discuss a "trend" in their results around line 254. This section needs to be reworded so that no trends are discussed. Journal policy around presentation of statistics do not permit such "trends" statements.

END OF COMMENTS

Dear Editor,

We appreciate your careful evaluation of our resubmission.

REQUIRED ITEMS

- You must upload original, uncropped western blot/gel images (including controls) if they are not included in the manuscript. This is to confirm that no inappropriate, unethical or misleading image manipulation has occurred. These should be uploaded as 'Supporting information for review process only'. Please label/highlight the original gels so that we can clearly see which sections/lanes have been used in the manuscript figures. For more information, see: <https://physoc.onlinelibrary.wiley.com/hub/journal-policies#imagnanip>.

We did not perform any western blot or gel analysis in this study.

Editor comment:

Thank you for the resubmission. We appreciate the new tables that summarize the statistics for many of the results. However, the authors still have not adequately addressed one of the concerns that I previously raised. They still discuss a "trend" in their results around line 254. This section needs to be reworded so that no trends are discussed. Journal policy around presentation of statistics do not permit such "trends" statements.

In response, we have carefully revised the sentences around line 254 and removed any mention of a "trend" from the Results section. The changes have been highlighted in yellow in the revised manuscript.

You have used SEM while the journal statistics policy required the use of SD. Please could you change this throughout and resubmit?

We have changed SEM to SD throughout our manuscript.

Dear Professor Yoshida,

Re: JP-RP-2025-288683R3 "Dual functions of SNAP25 in mouse taste buds" by Kengo Horie, Kuanyu Wang, Hai Huang, Keiko Yasumatsu, Yuzo Ninomiya, Yoshihiro Mitoh, and Ryusuke Yoshida

We are pleased to tell you that your paper has been accepted for publication in The Journal of Physiology.

Yours sincerely,

Nathan Schoppa
Senior Editor
The Journal of Physiology

If you would like to receive our 'Research Roundup', a monthly newsletter highlighting the cutting-edge research published in The Physiological Society's family of journals (The Journal of Physiology, Experimental Physiology, Physiological Reports, The Journal of Nutritional Physiology and The Journal of Precision Medicine: Health and Disease), please click this link, fill in your name and email address and select 'Research Roundup':
<https://www.physoc.org/journals-and-media/membernews>

- **TRANSPARENT PEER REVIEW POLICY:** To improve the transparency of its peer review process, The Journal of Physiology publishes online as supporting information the peer review history of all articles accepted for publication. Readers will have access to decision letters, including Editors' comments and referee reports, for each version of the manuscript as well as any author responses to peer review comments. Referees can decide whether or not they wish to be named on the peer review history document.
- You can help your research get the attention it deserves! Check out Wiley's free Promotion Guide for best-practice recommendations for promoting your work at: www.wileyauthors.com/eeo/guide. You can learn more about Wiley Editing Services which offers professional video, design, and writing services to create shareable video abstracts, infographics, conference posters, lay summaries, and research news stories for your research at: www.wileyauthors.com/eeo/promotion.
- **IMPORTANT NOTICE ABOUT OPEN ACCESS:** To assist authors whose funding agencies mandate public access to published research findings sooner than 12 months after publication, The Journal of Physiology allows authors to pay an Open Access (OA) fee to have their papers made freely available immediately on publication.

EDITOR COMMENTS

Thank you for addressing the final minor concerns that I had as Senior Editor. The manuscript has now been accepted.

I have one remaining request, which can be addressed in the Proofs stage. When you complete the proofs, please add a short sentence at the end of the Statistical Analysis section in the Methods indicating that the data in the figures are presented as Mean (SD). You have statements at the end of the legends for figures late in the manuscripts, but there are

earlier figures that present the data that way. This transparency will help readers.